# Geometric Embedding Alignment via Curvature Matching in Transfer Learning

## Abstract

Geometrical interpretations of deep learning models offer insightful perspectives into their underlying mathematical structures. In this work, we introduce a novel approach that leverages differential geometry, particularly concepts from Riemannian geometry, to integrate multiple models into a unified transfer learning framework. By aligning the Ricci curvature of latent space of individual models, we construct an interrelated architecture, namely Geometric Embedding Alignment via cuRvature matching in transfer learning (GEAR), which ensures comprehensive geometric representation across datapoints. This framework enables the effective aggregation of knowledge from diverse sources, thereby improving performance on target tasks. We evaluate our model on 23 molecular task pairs and demonstrate significant performance gains over existing benchmark models—achieving improvements of at least 14.4% under random splits and 8.3% under scaffold splits.

## 1 Introduction

Interest in the practical applications of deep learning has grown drastically over the years. Numerous examples have been announced recently, including applications in scientific domains such as biomedical, physical, and chemical sciences (Wang et al., 2019; Peng et al., 2021; Scarselli et al., 2009; Bruna et al., 2013; Duvenaud et al., 2015; Defferrard et al., 2016; Jin et al., 2018; Coley et al., 2019; Ko et al., 2023a;b; 2024; Yim et al., 2024; Lee et al., 2024). However, in most real-world application cases—regardless of the domain—the lack of data consistently poses a major obstacle. Considerable efforts have been devoted to overcoming this challenge. One promising approach involves leveraging transfer learning (TL) and multitask learning (MTL) to make use of information across different datasets, modalities, and tasks. (Zhuang et al., 2011; Long et al.; Zhuang et al., 2013; 2014; Pan et al., 2020; Quattoni et al., 2008; Kulis et al., 2011; Raghu et al., 2019; Yu et al., 2022)

TL, our primary focus, is a learning strategy that leverages information across different tasks to improve performance on a target task. Molecular property prediction tasks provide an excellent testbed for TL, as they typically involve relatively small datasets but a large number of prediction tasks per input molecule.

Most existing research has concentrated on classification tasks (Radhakrishnan et al., 2023; Basu et al., 2023; Wenzel et al., 2022), while relatively few approaches have been developed to support regression tasks—despite the fact that many practical applications in molecular sciences involve regression (Scarselli et al., 2009; Bruna et al., 2013; Duvenaud et al., 2015; Defferrard et al., 2016; Jin et al., 2018; Coley et al., 2019; Ko et al., 2023a;b; 2024; Yim et al., 2024; Lee et al., 2024). Given the real-world importance of regression problems, this underrepresentation is notable. Therefore, in this work, we focus on the regression-based TL setting applied to molecular property prediction and propose a novel method specifically tailored to this context.

By analyzing the general structure of TL, one can observe that there is always a 'bridging' component that connects different tasks to facilitate the flow of information. Our method redefines and enhances this bridging mechanism by reinterpreting the latent space as a smooth, curved geometry. Since a key aspect of TL is designing effective methods to couple tasks, this geometric viewpoint allows us to align tasks by directly matching the geometric properties of their latent spaces.

The fundamental approach of our novel method is based on Riemannian differential geometry. This is a reasonable hypothesis, as most deep learning models are constructed using smooth functions

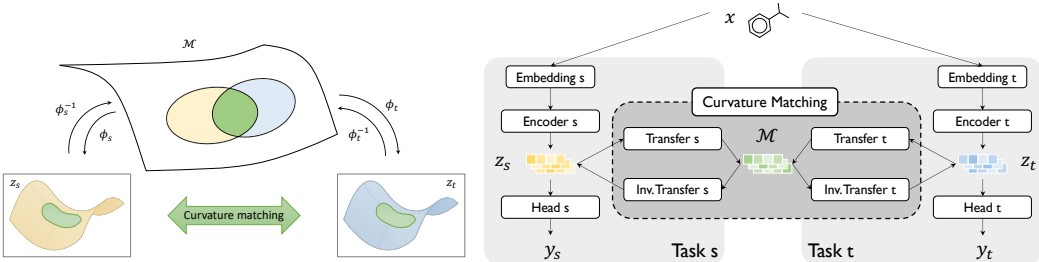

Figure 1: (Left) The framework consists of a common manifold $\mathcal{M}$, task-specific latent spaces $z_s$ and $z_t$, transfer functions $\phi_s$ and $\phi_t$. Their inverses, $\phi_s^{-1}$ and $\phi_s^{-1}$, map $z_s$ and $z_t$ to $\mathcal{M}$. (Right) Each task comprises five modules: embedding, encoder, transfer, inverse transfer, and head. Transfer and inverse transfer modules enable information exchange across tasks by curvature matching.

to ensure the feasibility of backpropagation. Consequently, the latent spaces produced by these models can also be considered smooth, being composed through the application of these smooth functions. Several studies in this field (Ko et al., 2023a;b; 2024; Yim et al., 2024; Lee et al., 2024) leverage the diffeomorphism invariance property of Riemannian geometry. These approaches have demonstrated effectiveness across multiple regression tasks in the molecular domain. However, despite their strengths, they also exhibit three major limitations inherent to their core algorithms.

One key limitation lies in the geometrical coverage of these algorithms. Because they operate by aligning infinitesimal distances between local perturbations, their effectiveness is inherently confined to local regions of the latent space. As a result, they struggle to capture the global geometric structure of the latent manifold. Furthermore, improving coverage typically requires increasing the number of perturbation points, which in turn leads to significant computational overhead.

Another important limitation is the potentially improper definition of 'infinitesimal.' In cases where the latent space of a task exhibits high curvature or warping, some perturbation points may no longer be validly considered infinitesimal. This undermines the core assumption of local linearity and can lead to inaccurate geometric alignment.

The final limitation lies in the necessity of a shared embedding layer across all tasks. To align infinitesimal distances between tasks, the perturbation points must be consistently defined within a shared latent space, which in turn requires a shared embedding layer. However, such a mechanism is often inadequate for handling inputs associated with different levels of complexity.

Hence, we propose a new model—Geometric Embedding Alignment via cuRvature matching (GEAR)—which extends the geometric foundations of GATE (Ko et al., 2023b) and broadens the scope of its geometric interpretation in TL. Unlike previous approaches that rely on the local boundary, our algorithm is built upon direct curvature matching, which, in turn, relaxes constraints on the input embedding structure. This allows for greater flexibility in customizing the model for individual tasks. We conducted regression experiments comparing GEAR to conventional TL methods using 23 pairs of molecular properties, and demonstrated that GEAR significantly outperforms them in most test cases. Furthermore, we validated the model's robustness through a series of ablation studies.

Our main contribution of the article is as follows.

- We design a novel TL algorithm GEAR based on Ricci curvature matching of latent spaces.

- GEAR significantly outperforms benchmark models in various molecular property regression tasks.

- GEAR exhibits stable geometry and robust behavior in extrapolation tasks.

## 2 RELATED WORKS

### 2.1 RIEMANNIAN GEOMETRY IN DEEP LEARNING

Geometric deep learning is a field that extends deep learning to non-Euclidean domains such as graphs and manifolds, gaining prominence for its ability to capture complex relational and structural patterns inherent in scientific, biological and real-world data(Bronstein et al., 2017). Riemannian differential geometry is a branch of mathematics that studies smooth manifolds equipped with a metric that allows the measurement of lengths and angles on the manifold. In the context of deep learning, this framework is instrumental in understanding and modeling the geometric structure of data, particularly in high-dimensional spaces. By treating data as lying on a manifold, Riemannian geometry facilitates the development of algorithms that respect the intrinsic geometry of the data, leading to more meaningful representations and improved performance in tasks such as classification(Pegios et al., 2024; Lee et al., 2022), clustering(Hu et al., 2024; Yang et al., 2018), and generative modeling(Park et al., 2023; Grattarola et al., 2019). Riemannian metric learning enhances deep learning by enabling models to operate in geometrically meaningful ways, improving interpretability and performance beyond Euclidean limits(Li et al., 2023; Sun et al., 2024).

### 2.2 TRANSFER LEARNING FOR MOLECULAR PROPERTY PREDICTION

TL has shown significant promise in molecular property prediction, particularly in data-scarce settings. (Falk et al., 2023) combine graph neural networks (GNNs) with kernel mean embeddings to enable knowledge transfer across atomistic simulations, capturing both local and global chemical features. (Buterez et al., 2024; Hoffmann et al., 2023) further extend this by leveraging multi-fidelity datasets, demonstrating that pretraining on low-fidelity data and fine-tuning on high-fidelity targets significantly improves molecular property prediction. (Yao et al., 2024) quantify task relatedness between molecular property prediction datasets, providing guidance for effective TL to enhance prediction performance.

In addition, recent studies have begun incorporating Riemannian differential geometry into TL frameworks for molecular property prediction. In (Ko et al., 2023b), source and target tasks are aligned by matching distances in infinitesimal regions of the latent space. The method is later generalized to a multi-task setup involving more than two tasks in (Ko et al., 2024). However, due to the computational burden of scaling this approach to many tasks, (Yim et al., 2024) introduce a task addition strategy to accelerate training.

## 3 METHODS

A geometric interpretation of latent space requires some mathematical preliminaries. The appropriate mathematical framework for describing curved spaces is differential geometry. Therefore, we briefly introduce the fundamental definitions and expressions that will be used in the following sections, along with the core ideas underlying our proposed method. (Check Appendix B for more details)

Since deep learning models always have a smooth underlying structure due to the backpropagation algorithm, it is very natural to assume that the latent space forged by a model is also smooth. Hence, it is plausible to assume the space is Riemannian. Detailed logical justifications for this assumption are provided in Appendix B.1.

Let us consider the space in which the input dataset resides. This space contains all the information that can be utilized to perform any kind of downstream task. When a specific downstream task is fixed, the latent space formed by the downstream model effectively retracts the original space into a smaller, task-specific subspace to enhance performance. However, since the latent vectors originate from the same universal input space, the latent vector corresponding to a different downstream task should also represent the same point in that universal space. To reconcile latent representations from different downstream models, we leverage diffeomorphism invariance to construct an intermediate space with a locally flat frame, allowing us to align latent vectors from distinct downstream tasks.

Now, the real question is: how? In previously published methods (Ko et al., 2023b; 2024; Yim et al., 2024), a perturbation strategy is used to align task-specific spaces. However, this approach has several notable drawbacks such as limited coverage of geometries and the requirement of a shared embedding

layer. To address these issues, we extend the underlying idea by aligning the geometries of latent spaces through the matching of Ricci curvatures computed from each space. Since the computation of the Ricci scalar is highly intricate, we first introduce the basic forms of its constituent elements here, and provide a more detailed theoretical and mathematical walkthrough in the Appendix B and C.

## 3.1 PRELIMINARY

To maintain abstract notation, we will use the Einstein summation convention with index contraction representation. A fundamental introduction to these concepts is provided in the Appendix A.

Riemannian geometry is often characterized by the Ricci scalar curvature. To understand how curvature is induced, one must carefully follow a step-by-step calculation process.

Everything begins with the metric. A metric is a rank-2 tensor with a symmetric property, which is crucial for computing distances between two points on a curved space. However, there is no systematic method to derive the explicit form of the metric for a given space directly. Instead, one must rely on a key mathematical property of Riemannian manifolds.

A Riemannian manifold always guarantees diffeomorphism invariance—in other words, freedom in the choice of coordinates on the manifold. This allows for the existence of a locally flat coordinate system under any circumstance. In such a system, the metric can be induced from the flat metric $\eta_{ij}$ by applying the Jacobian of the coordinate transformation at a given point. Here, $x'^i$ and $x^i$ are points on curved and locally flat frame respectively, and then, the Jacobian of the transformation between these coordinates then takes the following form.

$$J^i{}_j = \frac{dx'^i}{dx^j} \tag{1}$$

From the Jacobian $J^i{}_j$, one can compute the induced metric in a straightforward manner.

$$g_{ij} = \frac{dx'^m}{dx^i} \eta_{mn} \frac{dx'^n}{dx^j} = \frac{dx'^m}{dx^i} \frac{dx'_m}{dx^j} \tag{2}$$

By obtaining the curved metric $g_{ij}$, one can define the Christoffel symbols $\Gamma^i{}_{jk}$, which are used to construct the covariant derivative $\nabla_i$ —replacing the ordinary derivative in Riemannian geometry.

$$\Gamma^i{}_{jk} = \frac{1}{2} g^{im} (\partial_j g_{mk} + \partial_k g_{mj} - \partial_m g_{kj}) \tag{3}$$

And the covariant derivative takes the following form.

$$\nabla_j T^i = \partial_j T^i + \Gamma^i{}_{jl} T^l \tag{4}$$

The curvature of a manifold $R^i{}_{ljk}$ can be defined by the difference between tangent vectors that are parallel transported along different paths from the same initial point to the same final point. This can be expressed using the commutation relation of two covariant derivatives acting on a tangent vector.

$$R^i{}_{ljk} T^l = [\nabla_j, \nabla_k] T^i \tag{5}$$

Finally, by contracting $i$ and $j$, and $l$ and $k$ respectively, the Ricci scalar curvature $R$ can be obtained.

$$R = g^{ij} g^{lk} R_{iljk} \tag{6}$$

The scalar curvature is invariant under diffeomorphisms, as is evident from the fact that it has no free indices. Consequently, this quantity is often used to characterize the curvature of a given manifold. Since we are working with curved latent spaces and aiming to connect two different curved coordinate representations originating from a universal curved manifold, we directly compute this scalar property and align it to match the shapes of the task-specific spaces.

## 3.2 ANALYTIC COMPUTATION STRATEGY

A deep learning model is composed of multiple smooth layers. Therefore, if differentiable activation functions are used, it becomes possible to compute the curvature tensor of the curved space induced by the model. However, when the model consists of many layers, it becomes convenient to define

building blocks that allow the full Jacobian to be computed by simply multiplying them. These building blocks can be expressed in terms of the weights and biases of each layer. Starting from the full Jacobian, and by applying the chain rule, the Jacobian can be decomposed into the Jacobians of individual layers.

$$J^i_j = \frac{dx'^i}{dx^j} = \frac{dx'^i}{dx^{(n-1)k_{n-1}}} \frac{dx^{(n-1)k_{n-1}}}{dx^{(n-2)k_{n-2}}} \cdots \frac{dx^{(1)k_1}}{dx^j} \tag{7}$$

Here, $n$ denotes the layer index of the transfer module in the model, as illustrated in Figure 1. Therefore, when similar mathematical structures appear across layers—as is often the case—it becomes possible to define a fundamental building block of the full Jacobian using the Jacobian of a single representative layer. In our setup, each layer follows a linear MLP structure with the SiLU activation function. The fundamental Jacobian block can then be expressed in the following form:

$$
\begin{aligned}
\tfrac{dx^{(n+1)i}}{dx^{(n)j}} =\quad & W^{(n+1)i}{}_k(((x^{(n)k})e^{-x^{(n)k}} \times \mathrm{LS}(x^{(n)k}) + 1)\mathrm{LS}(x^{(n)k}))^k{}_j \\
=\quad & (W^{(n+1)i}{}_j\sigma^i + (W^{(n+1)}x^{(n)} + b^{(n+1)})^i W^{(n+1)a_3}{}_j E^i{}_{a_3}(\sigma^2)^i)
\end{aligned}
\tag{8}
$$

Here, $W^{(n)i}{}_j$ and $b^{(n)i}$ are weights and biases of $n$-th layer in the transfer module. The new notations introduced in the equation above are defined as follows. First, $\mathrm{LS}(x)$ denotes the logistic function and $\sigma^i$ and $E^i{}_l$ are expressed as follows:

$$
\sigma^i = \frac{1}{1 + e^{-(W^{(n)i}{}_j x^j + b^{(n)i})}}, \qquad
\mathrm{E}^i{}_l \equiv (e^{-(W^l{}_j x^j + b^j)})^i{}_l = \begin{cases} e^{-(W^l{}_j x^j + b^l)} & \text{if } l = i \\ 0 & \text{if } l \neq i \end{cases}
\tag{9}
$$

$(\sigma^2)^i$ denotes the element-wise square of $\sigma^i$. By utilizing Eq. 8, it is now possible to compute the full Jacobian of the transfer module. The induced metric can then also be specified by Eq. 2.

However, this is not sufficient to compute the curvature. To express curvature explicitly in terms of the metric, two additional components are required: the first derivative of the metric tensor—since the Christoffel symbols are defined using both the metric and its derivatives—and the second derivative of the metric tensor, as curvature depends on the derivatives of the Christoffel symbols. Therefore, we need to identify two additional fundamental building blocks to compute the curvature tensor. The first derivative of the metric tensor can be expressed as a combination of the Jacobian and the derivative of the Jacobian. Thus, the next step is to derive the explicit form of the Jacobian's derivative.

$$
\begin{aligned}
\tfrac{\partial^2 x^{(n+1)i}}{\partial x^{(n)k} \partial x^{(n)j}} =\quad & W^{(n+1)i}{}_j W^{(n+1)a_2}{}_k E^i{}_{a_2}(\sigma^2)^i + W^{(n+1)i}{}_k W^{(n+1)a_3}{}_j E^i{}_{a_3}(\sigma^2)^i \\
& -(W^{(n+1)}x^{(n)} + b^{(n+1)})^i W^{(n+1)a_3}{}_j W^{(n+1)i}{}_k E^i{}_{a_3}(\sigma^2)^i \\
& +2(W^{(n+1)}x^{(n)} + b^{(n+1)})^i W^{(n+1)a_3}{}_j E^i{}_{a_3} W^{(n+1)a_6}{}_k E^i{}_{a_6}(\sigma^3)^i
\end{aligned}
\tag{10}
$$

Finally, the derivative of the Christoffel symbols can be induced with the second derivative of the Jacobian.

$$
\begin{aligned}
\tfrac{\partial^3 x^{(n+1)i}}{\partial x^{(n)l} \partial x^{(n)k} \partial x^{(n)j}} =\ & \\
& -2W^{(n+1)i}{}_l W^{(n+1)a_2}{}_k W^{(n+1)i}{}_j E^i{}_{a_2}(\sigma^2)^i - W^{(n+1)i}{}_j W^{(n+1)a_3}{}_l W^{(n+1)i}{}_k E^i{}_{a_3}(\sigma^2)^i \\
& +4W^{(n+1)i}{}_l W^{(n+1)a_2}{}_k W^{(n+1)a_9}{}_j E^i{}_{a_2} E^i{}_{a_9}(\sigma^3)^i \\
& +(W^{(n+1)}x^{(n)} + b^{(n+1)})^i W^{(n+1)a_3}{}_l W^{(n+1)i}{}_k W^{(n+1)i}{}_j E^i{}_{a_3}(\sigma^2)^i \\
& -2(W^{(n+1)}x^{(n)} + b^{(n+1)})^i W^{(n+1)a_3}{}_l W^{(n+1)i}{}_k W^{(n+1)a_9}{}_j E^i{}_{a_3} E^i{}_{a_9}(\sigma^3)^i \\
& +2W^{(n+1)i}{}_j W^{(n+1)a_3}{}_l W^{(n+1)a_6}{}_k E^i{}_{a_3} E^i{}_{a_6}(\sigma^3)^i \\
& -4(W^{(n+1)}x^{(n)} + b^{(n+1)})^i W^{(n+1)a_3}{}_l W^{(n+1)a_6}{}_k W^{(n+1)i}{}_j E^i{}_{(a_3} E^i{}_{a_6)}(\sigma^3)^i \\
& +6(W^{(n+1)}x^{(n)} + b^{(n+1)})^i W^{(n+1)a_3}{}_l W^{(n+1)a_6}{}_k W^{(n+1)a_9}{}_j E^i{}_{a_9} E^i{}_{a_3} E^i{}_{a_6}(\sigma^4)^i
\end{aligned}
\tag{11}
$$

By gathering and utilizing the three building blocks described above and imposing them into Eq. 2, 3, 5 and 6 , the scalar curvature of the given curved space can be explicitly calculated.

## 3.3 MODEL ARCHITECTURE

Our model is designed to perform effectively in a two-task setting, regardless of whether the data distributions between tasks are balanced or unbalanced. Therefore, the basic architecture consists of

two distinct task-specific models connected by a transfer module, as shown in Figure 1. These task-specific models are connected only through the curvature matching section; thus, their architectures are fully flexible, with the sole constraint that the dimensions of the latent vectors fed into the transfer module must match. This allows each task-specific model to be independently designed, taking into account the complexity of the task and its corresponding data distribution.

When an input data point is fed into the model, the first step is to construct an embedding vector from the input information. We denote the embedding vector as $z_t$ for the target task and $z_s$ for the source task. These embedding vectors follow two distinct paths in the architecture: one path leads to the transfer module, and the other proceeds to the head module in the model. The transfer module maps each embedding to a vector of the same dimension in a locally flat coordinate frame.

$$z' = \text{Tran}(z), \qquad \hat{z} = \text{Tran}^{-1}(z') \tag{12}$$

However, for the inverse transfer, direct computation of the inverse matrix during backpropagation can be unstable. To address this, we designed an autoencoder architecture to map the embedding vector from the locally flat frame back to the original space. Accordingly, we define an autoencoder loss to guide this reconstruction process.

$$l_{\text{auto}} = \text{MSE}(z, \hat{z}) \tag{13}$$

Since the transferred vectors $z'_s$ and $z'_t$ originate from the same input, they should match—assuming the coordinate systems are aligned, i.e., expressed in a common locally flat frame.

$$z'_s = \text{Tran}_{s \to LF}(z_s), \quad \hat{z}_s = \text{Tran}^{-1}_{LF \to s}(z'_s), \quad z'_t = \text{Tran}_{t \to LF}(z_t), \quad \hat{z}_t = \text{Tran}^{-1}_{LF \to t}(z'_t) \tag{14}$$

Here, $\text{Model}_{s \to LF}$ denotes the transfer module that maps the embedding vector from the source space to the locally flat (LF) frame, and vice versa. To encourage alignment, we introduce a consistency loss by matching the embedding vectors from both the source and target task models within the shared locally flat frame.

$$l_{cons} = \text{MSE}(z'_s, z'_t) \tag{15}$$

To further reinforce the connection between the source and target tasks, we introduce an additional loss—the mapping loss—which aligns the downstream prediction of the original target label with the prediction obtained from an embedding vector transferred from the source model.

$$l_{map} = \text{MSE}(y_t, \hat{y}_{s \to t}) \tag{16}$$

Here, $y_t$ denotes the target label and $\hat{y}_{s \to t}$ represents the predicted value obtained from the embedding vector transferred from the source model. And the ordinary regression loss is also important.

$$l_{reg} = \text{MSE}(y_t, \hat{y}_t) \tag{17}$$

Finally, we define the curvature and metric losses. The metric loss plays a crucial role, as the space formed by the transfer module lacks any form of direct supervision. Without proper regularization, the space is not guaranteed to be locally flat, since there are infinitely many ways to define a basis that still satisfy the previously introduced constraints. The metric loss guides the transfer module toward preserving local flatness. It is defined as the discrepancy between the induced flat metric and the Euclidean metric, which in this case is represented by the identity matrix $\eta_{ij}$.

$$l_{metric} = \text{MSE}(\eta_{ij}, \eta_{(s)_{ij}}) + \text{MSE}(\eta_{ij}, \eta_{(t)_{ij}})$$

$$\eta_{(s)ij} = \left(\frac{\partial \hat{z}_s^m}{\partial \hat{z}_s'^i}\right)\left(\frac{\partial z_s'^k}{\partial \hat{z}_s^m}\right)\eta_{kl}\left(\frac{\partial z_s'^l}{\partial \hat{z}_s^n}\right)\left(\frac{\partial \hat{z}_s^n}{\partial \hat{z}_s'^j}\right), \quad \eta_{(t)ij} = \left(\frac{\partial \hat{z}_t^m}{\partial \hat{z}_t'^i}\right)\left(\frac{\partial z_t'^k}{\partial \hat{z}_t^m}\right)\eta_{kl}\left(\frac{\partial z_t'^l}{\partial \hat{z}_t^n}\right)\left(\frac{\partial \hat{z}_t^n}{\partial \hat{z}_t'^j}\right) \tag{18}$$

$\eta_{(s)ij}$ and $\eta_{(t)ij}$ denote the induced flat metrics obtained through iterative $K$ loop computations using the inverse transfer mappings from the source and target, respectively, into the transfer module. This back-and-forth mapping between the task coordinate and the locally flat coordinate can be repeated $K$ times. As the loop proceeds, the metric is repeatedly transformed under the diffeomorphism, and error accumulates. Increasing $K$ therefore imposes a stronger constraint on the metric but also amplifies its sensitivity. For this reason, we keep $K$ small in the setup. (see Algorithm 1 and Appendix E)

The final loss term is the *curvature matching loss*. Since we have already introduced the fundamental building blocks for computing scalar curvature in Sections 3.1 and 3.2, the scalar curvature can now

be computed analytically. Given that the curvatures of the target and source spaces should align, we define this curvature matching loss as the most critical and final component of our architecture.

$$l_{curv} = \text{MSE}(R_s, R_t) \tag{19}$$

Where $R_t$ and $R_s$ are the Ricci scalar curvatures from the target and the source respectively. Finally, by combining all with appropriate hyperparameters, the main loss of the model can be defined.

$$l_{tot} = l_{reg} + \alpha l_{auto} + \beta l_{cons} + \gamma l_{map} + \delta l_{metric} + \epsilon l_{curv} \tag{20}$$

Each hyperparameter can be tuned individually to improve the model's predictive performance. In particular, the weight of the metric loss often needs to be increased, as its raw magnitude is substantially smaller than that of the other loss terms. The specific configurations of these loss components and the associated model parameters are described in Appendix E, with an ablation study on hyperparameter tuning provided in Appendix H.2. In addition, detailed schematics of our model are shown in Appendix Figure 6.

## 4 EXPERIMENTS

### 4.1 EXPERIMENTAL SETTINGS

The experiments are conducted using open datasets from three distinct databases (OCHEM (Sushko et al., 2011), PubChem (Kim et al., 2022), and CCCB (III, 2022)) forming 23 task pairs across 14 different tasks using two distinct data splitting schemes: the conventional random split and the scaffold-based split, the latter of which simulates OOD scenarios. We used directional message passing network (DMPNN) (Yang et al., 2019) for encoding molecular structures. A detailed explanation of these datasets and their corresponding prediction tasks is provided in the Appendix F. To ensure the robustness of the results, all experiments are performed using 4-fold cross-validation. Each experiment is run on a single NVIDIA A40 GPU.

To evaluate our method, we compare it against several benchmark models, including single-task learning (STL), MTL, global structure preserving loss-based knowledge distillation (GSP-KD) (Joshi et al., 2022), two variants of TL (retraining all layers vs. retraining the head only), and GATE (Ko et al., 2023b). We ensure fairness by maintaining the same encoder and head architectures across all benchmark models and our method. Detailed backbone architecture and hyperparameter configurations are in the Appendix D and E.

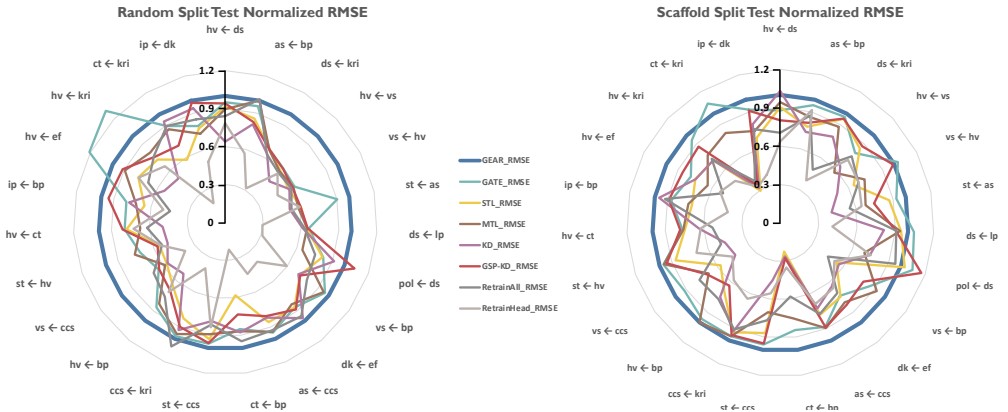

Figure 2: The results are illustrated in the form of a radar chart. Each axis plots the GEAR RMSE divided by the benchmark model RMSE. The baseline in the chart corresponds to the RMSE performance of GEAR, which is normalized to 1. (higher is better). Due to space constraints, the detailed experimental results are provided in Table [5, 6, 7, 8] in the Appendix G

### 4.2 MAIN RESULTS

Figure 2 demonstrates the superior performance of our algorithm compared to other benchmark models. In both data split schemes, our model consistently outperforms the baseline models by

considerable margins. Notably, when counting the number of best-performing tasks, GEAR achieves the lowest RMSE in 18 out of 23 task pairs under the random split and in 17 out of 23 under the scaffold split. Furthermore, when including second-best performances, GEAR ranks within the top two in 22 out of 23 for both split schemes.

From a performance standpoint, GEAR improves the average RMSE over GATE by $14.4\%$ in the random split and by $8.3\%$ in the scaffold split. Compared to the third-best model, GEAR achieves an improvement of $22.8\%$ (MTL) under the random split and $21.4\%$ (GSP-KD) under the scaffold split.

# 5 ABLATION STUDIES

## 5.1 ROLE OF CURVATURE LOSS

Since GEAR is constructed under a TL scheme, it is crucial to verify that the loss terms connecting the source and target tasks effectively facilitate information transfer. To support this claim, we conducted three different experiments and plotted training and validation accuracy curves.

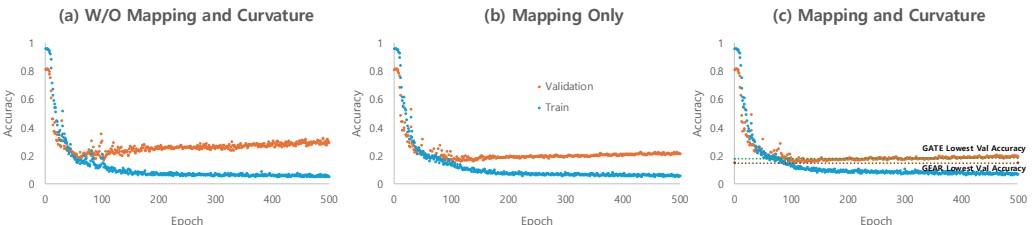

Figure 3: These plots illustrate the primary role of the curvature loss tested on dk $\rightarrow$ hv task pair. In figure (a), both the mapping loss and the curvature loss are turned off. In figure (b), only the mapping loss is enabled. In figure (c), both the mapping and curvature losses are activated.

As shown in Figure 9, when both the mapping and curvature matching losses are turned off, the loss curve exhibits a severe overfitting issue. Enabling the mapping loss alone helps to stabilize this overfitting to some extent. However, when both losses are activated, overfitting is significantly suppressed, and the validation accuracy reaches the lowest value overall.

For comparison, we included the minimum validation value of GATE as a green dotted line, alongside that of GEAR (in brown dotted line) under the same experimental setting. The comparison reveals that GEAR achieves a lower minimum validation than GATE, with an improvement margin of $17.5\%$.

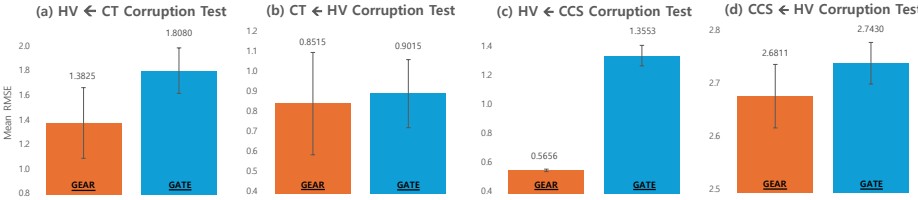

Figure 4: This figure highlights the performance of GEAR on corrupted data, demonstrating that it either outperforms or performs comparably to GATE. The values represent the average RMSE across four folds, with the STD error bars. Specifically, Figure (a) shows HV prediction results using CT as the source, (b) shows CT prediction results using HV as the source, (c) shows HV prediction results using CCS as the source, and (d) shows CCS prediction results using HV as the source.

## 5.2 ROBUSTNESS UNDER CORRUPTED DATASET

In this subsection, we demonstrate the robustness of GEAR under targeted corruption stress on dataset to assess its regularization effect. We corrupted data points with values at least twice the standard deviation of each dataset. Specifically, we selected $10\%$ of the test set containing values greater than

the dataset's standard deviation. These selected labels were corrupted by multiplying them by -1 and then injected into the training dataset. This setup reflects common scientific data errors, such as missing minus signs or inconsistent units. After training, we evaluated the model by feeding these corrupted samples and comparing the predictions against their original, uncorrupted labels. This procedure was repeated under a 4-fold cross-validation scheme to ensure the reliability.

As shown in Figure 4, GEAR consistently outperforms conventional models across all cases.

## 5.3 Computational costs

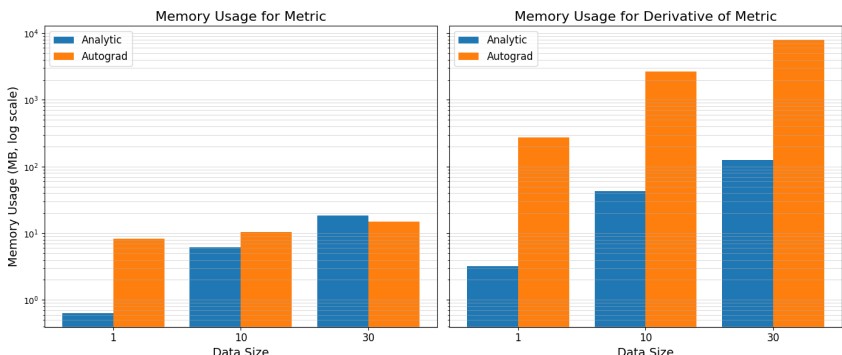

Figure 5: Memory usage was visualized in the form of bar charts in log scale. The charts compare the memory consumption of the analytic and autograd-based methods when computing the metric and the derivative of the metric, evaluated at data sizes of 1, 10, and 30. The corresponding bars are labeled as Analytic and Autograd, respectively.

In this subsection, we demonstrate the necessity of computing the curvature analytically. Although metric and its derivatives can be calculated by autograd, autograd requires substantial computational cost compared to analytic computation. We compared memory consumption between the analytical and autograd-based methods when computing the metric and its derivatives across varying data sizes.

As shown in Figure 5, both methods exhibited similar memory usage for metric computation, which involves first-order derivatives. However, for computing metric derivatives (i.e., second-order), the autograd approach consumed approximately 85.5× more memory at data size 1. Due to this overhead, autograd-based training was infeasible under our GPU constraints. In contrast, the analytical method enabled fast and memory-efficient training, requiring only 0.5 seconds per iteration at batch size 512.

## 6 Discussion

We introduced a novel TL algorithm, GEAR, based on Riemannian differential geometry. Since deep learning models are inherently smooth and differentiable, the Jacobian of the transfer module can be computed analytically. From the Jacobian, the induced curved metric can be derived and used for curvature computation. The Ricci scalar curvature encapsulates the full geometric characteristics of the latent space; by matching the curvature between the target and source tasks, the latent spaces can be accurately aligned. Experimental results on 23 pairs of molecular property prediction tasks demonstrate the superior performance of GEAR compared to benchmark models.

Simplifying or relaxing the curvature matching process—without sacrificing generality—helps reduce the implementation complexity and computational overhead typically associated with curvature computation. GEAR also introduces structural flexibility by connecting source and target tasks through transfer modules, without imposing restrictions on the downstream architecture. This allows the encoder modules to remain fully unconstrained, enabling seamless adaptation to multi-modal learning scenarios. Moreover, the framework is inherently extensible to settings involving more than two interrelated tasks, opening opportunities for broader applications in multi-task transfer learning. Importantly, these extensions are not limited to the chemical domain, and can be applied to other areas such as natural language processing (NLP) and computer vision.

# 7 REPRODUCIBILITY STATEMENT

Due to patent considerations, we cannot release source code in the supplementary materials. However, we provide comprehensive descriptions of the model architecture (Figures 1, 6), the full set of equations (Sections 3, C), and pseudo-code (Algorithm 1). Hyperparameters and dataset details are given in Section F, while background material on differential geometry is summarized in Sections A and B to support readers who are less familiar with this area. Together, these resources should enable reproducibility of our results.

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

## A  NOTATIONS

Our notation follows index notation and the Einstein summation convention. The functions and matrices used in our algorithm are defined as follows.

$$X : \text{Vector} \tag{21}$$
$$X^\mu : \text{Vector Field} \tag{22}$$
$$dx_\mu : \text{Basis} \tag{23}$$
$$X_\mu : \text{Dual Vector Field} \tag{24}$$
$$dx^\mu : \text{Dual Basis} \tag{25}$$
$$T : \text{Tensor} \tag{26}$$
$$T^{\nu_1 \cdots \nu_p}{}_{\mu_1 \cdots \mu_q} : \text{(p, q) Tensor Field} \tag{27}$$
$$g_{\mu\nu} : \text{Metric Tensor} \tag{28}$$
$$\delta_{\mu\nu} : \text{Kronecker Delta} \tag{29}$$
$$\nabla_\mu : \text{Covariant Derivative} \tag{30}$$
$$\mathcal{L}_X : \text{Lie Derivative} \tag{31}$$
$$\Gamma^\rho{}_{\mu\nu} : \text{Christoffel Symbol} \tag{32}$$

All indices are raised and lowered by the metric $g_{\mu\nu}$. For instances,

$$g^\mu{}_\nu = g^{\mu\rho} g_{\rho\nu} \tag{33}$$

where

$$g^{\mu\nu} g_{\mu\nu} = \delta^\mu{}_\nu = D \tag{34}$$

Here $D$ is the number of dimensions.

## B  MOTIVATIONS AND THEORETICAL BACKGROUNDS

We prepared this section to assist readers who may not be familiar with the mathematical foundations of differential geometry. The content is essentially a summarized compilation of well-established textbook materials, including (Weinberg, 1972; Carroll, 2004; Wald, 1984). In addition, the material in subsections B.3, B.5, and B.6 is also covered in the original GATE paper (Ko et al., 2023b).

### B.1  JUSTIFICATION FOR THE RIEMANNIAN GEOMETRY ASSUMPTION

In this subsection, we clarify the motivations and the rationale behind this assumption and why it is both theoretically sound and practically justified in the context of deep learning. While this was briefly mentioned in Section 3 of our manuscript and supported by citations in Section 2, we acknowledge that a more explicit theoretical justification is warranted. Below, we provide a detailed rationale to clarify why this assumption is both mathematically valid and practically appropriate in the context of our method.

The assumption that latent spaces in deep learning models can be treated as Riemannian manifolds rests on the following logical reasoning:

1. A **Riemannian manifold** is formally defined as a *smooth manifold* equipped with a *Riemannian metric*—a smoothly varying inner product on each tangent space.

2. A classical result in differential geometry establishes that **any smooth manifold admits a Riemannian metric**. This is a well-known theorem found in standard references such as (Lee, 2018)

3. The critical question, then, is whether the latent space of a deep learning model qualifies as a smooth manifold. This can be affirmed based on the construction of modern deep neural networks:

- *Linear transformations* are inherently smooth mappings.
- *Nonlinear activation functions* (e.g., Tanh, Sigmoid, SiLU) are continuously differentiable and thus smooth.

4. Therefore, when smooth activation functions are used, the entire model becomes a composition of smooth functions. The resulting latent space—formed by mappings from the input through the network—is itself smooth and hence forms a smooth manifold.

Based on the above, it follows directly that the latent space **can be equipped with a Riemannian metric**, rendering it a Riemannian manifold.

This assumption not only holds mathematically but is also aligned with practices in prior literature, including GATE and other geometric learning frameworks. Modeling latent spaces as Riemannian manifolds enables the use of powerful geometric tools—such as curvature—to capture structural properties that are otherwise inaccessible through Euclidean assumptions. In our case, this motivates the introduction of Ricci curvature alignment as a principled approach to improving transfer performance.

## B.2 MOTIVATION FOR RICCI CURVATURE MATCHING

Our work builds on the GATE architecture (Ko et al., 2023b) and thus inherits foundational assumptions from that framework. As described in the GATE paper, each data point within a task lies on a manifold, and the set of such points forms a coordinate patch, interpretable as a task-specific coordinate system. This is reasonable because many downstream tasks originate from a universal molecular representation (e.g., SMILES), with task-specific latent representations viewed as coordinate transformations of the same underlying structure.

Given the latent space's smoothness, it can be modeled as a Riemannian manifold (as argued in the previous section). Accordingly, task-specific latent spaces can be connected via diffeomorphisms—smooth, invertible mappings between manifolds.

We require the following assumptions to hold for the dataset:

- The source and target tasks are correlated.
- Their distributions share overlapping regions.

These assumptions are realistic, as our dataset includes many scientifically correlated task pairs, and most molecules have multiple annotated properties.

The fundamental strategy in both GATE and GEAR is to use source-task data to compensate for underrepresented regions in the target task. Given a Riemannian latent space, we can transfer knowledge across tasks by learning diffeomorphic mappings between latent representations.

To make this concrete, consider the following example: Suppose we have two molecules—water and oil. We know the melting point of water but not its boiling point; for oil, we have both values. If boiling point prediction is the target task and melting point is the source task, then we can train the model to learn a mapping from oil's melting-point representation to its boiling-point representation. Once trained, the model can infer water's boiling-point representation from its melting-point latent vector, transferring knowledge via the learned transformation. This enables improved performance on the target task.

Understanding the geometry of latent spaces is essential for meaningful transfer. Riemannian manifolds are inherently curved, and standard derivatives are insufficient for accurately describing vector displacement. Instead, one must use the *covariant derivative*, which accounts for curvature through the Christoffel symbol. This term varies across coordinate systems, making it equivariant rather than invariant.

The second derivative of the metric leads to the *curvature tensor*, which characterizes the manifold's intrinsic geometry. Among its contractions, the *Ricci scalar* is of particular interest: it is diffeomorphism-invariant and summarizes the manifold's curvature using a single scalar value.

Since all Riemannian manifolds enjoy diffeomorphism invariance, this property provides freedom in coordinate choice. Formally, a diffeomorphism is a smooth bijective map with differentiable inverse.

Practically, it means that a vector's intrinsic properties remain unchanged even when expressed in a new coordinate basis. Consequently, one can always find a coordinate frame in which the manifold appears locally flat.

To uncover the latent manifold's geometry, one could:

- Solve the Einstein field equations to obtain the metric tensor.
- Propose a suitable *Ansatz* and verify that it satisfies the Einstein equations.
- Define a mapping function and derive the curved metric from a flat one using the Jacobian.

The third method is the most practical in deep learning. General solutions to Einstein's equations are unknown for arbitrary settings, and crafting a good *Ansatz* is difficult and task-dependent. However, the Jacobian-based approach is well-established: the curved metric is computed from the Jacobian and its inverse, composed with a flat metric.

Using this method, we analytically compute curvature for task-specific manifolds and compare their geometry through Ricci scalars.

We chose to extend GATE by replacing local perturbation alignment with Ricci curvature matching. As discussed in our Introduction, this provides several key benefits:

- It captures **global geometric structure**, rather than relying on limited local perturbations.
- It removes ambiguity in choosing "infinitesimal" scales—especially relevant when latent vector magnitudes vary or curvature is large.
- It eliminates the need for perturbation-based sampling and supports a **universal embedding space** that enables non-linear mappings and potential multi-modal extensions.

In summary, Ricci curvature offers a mathematically principled, empirically effective, and computationally viable means of aligning task-specific latent spaces in transfer learning.

## B.3 THE DEFINITION OF RIEMANNIAN MANIFOLD

A curved space is complicated to comprehend in general. Since the late 19th century, there has been immense development in differential geometry to formally interpret curved spaces. One of the best-known intuitive geometries is Riemannian geometry. Riemannian geometry possesses a handful of useful mathematical properties that can be utilized in the real world. The formal definition of Riemannian geometry is as follows:

**Definition B.1** (Riemannian Manifold). A Riemannian metric on a smooth manifold M is a choice at each point $x \in M$ of a positive definite inner product $g_p : T_pM \times T_pM \to \mathbb{R}$ on $T_xM$. The smooth manifold endowed with the metric $g$ is a Riemannian manifold, denoted $(M, g)$.

As stated above, a Riemannian manifold is smooth and differentiable everywhere on the manifold, along with its derivatives. Moreover, a Riemannian manifold enjoys diffeomorphism invariance, induced by the Lie derivative $\mathcal{L}_X$. It can be readily observed that the composition of two different Lie derivatives forms a group, known as the diffeomorphism group. This isometry guarantees that coordinate choices can be made without altering the global geometry of the space.

$$X' = X'^\mu dX'_\mu = X'^\mu \frac{\partial X^\nu}{\partial X'^\mu} dX_\nu = X^\nu dX_\nu = X \tag{35}$$

As shown in Eq. 35, the transformed vector remains unchanged. Moreover, it is always possible to fix the transformed coordinates in a locally flat space.

$$\xi^\mu = \frac{\partial \xi^\mu}{\partial X^\nu} X^\nu \tag{36}$$

Where $\xi^\mu$ is a vector on a locally flat frame. To ensure the vector is on a flat frame, one must impose the following condition:

$$\frac{\partial^2}{\partial t^2} \xi^\mu(t) \equiv 0 \tag{37}$$

Since a vector is on a flat frame, it should be in free-falling motion, and thus its acceleration should be trivial. On a locally flat frame, the metric also reduces to the flat Euclidean metric.

$$g_{\mu\nu} = 1_{\mu\nu} \tag{38}$$

B.4 COVARIANCE

A vector should transform consistently across any coordinate frame. However, if the space is no longer flat, the ordinary derivative no longer preserves this property. To address this, let us consider the derivative of a vector in a general curved space.

$$\partial_\mu \to \partial'_\mu = \frac{\partial x^\mu}{\partial x'^\nu}\partial_\nu \tag{39}$$

Where $\partial_\mu = \frac{\partial}{\partial x^\mu}$, the vector transformation can be written as follows:

$$\partial_\nu X^\mu \to \partial'_\nu X'^\mu = \frac{\partial x^\lambda}{\partial x'^\nu}\frac{\partial}{\partial x^\lambda}\left(\frac{\partial x'^\mu}{\partial x^\rho}V^\rho\right) \tag{40}$$

$$= \frac{\partial x'^\nu}{\partial x^\lambda}\left(\frac{\partial x'^\rho}{\partial x^\nu}\partial^\lambda V^\rho + \frac{\partial^2 x'^\mu}{\partial x^\lambda \partial x^\rho}V^\rho\right) \tag{41}$$

As shown above, the transformation of a vector on a curved space using an ordinary derivative is no longer covariant. Therefore, it is necessary to introduce an additional term to restore covariance, namely the affine connection. With this addition, one can define the covariant derivative, which replaces the ordinary derivative.

$$\nabla_\mu = \partial_\mu + \Gamma^\lambda{}_{\mu\nu} \tag{42}$$

By imposing the covariance condition on the covariant derivative,

$$\nabla_\lambda \to \nabla'_\lambda V'^\mu = \frac{\partial x^\rho}{\partial x'^\nu}\frac{\partial x'^\mu}{\partial x^\nu}\nabla_\rho V^\nu \tag{43}$$

one can derive the explicit form of the connection.

$$\nabla_\mu V^\nu = \partial_\mu V^\nu + \Gamma^\nu{}_{\mu\lambda}V^\lambda \tag{44}$$

Under coordinate transformation,

$$\frac{\partial}{\partial x'^\mu}\left(\frac{\partial x'^\nu}{\partial x^\lambda}V^\lambda\right) + \Gamma'^\nu{}_{\mu\sigma}V'^\sigma = \frac{\partial x^\rho}{\partial x'^\mu}\frac{\partial x'^\nu}{\partial x^\lambda}\partial_\rho V^\lambda + \frac{\partial x^\rho}{\partial x'^\mu}\frac{\partial^2 x'^\nu}{\partial x^\rho \partial x^\lambda}V^\lambda + \Gamma'^\nu{}_{\mu\sigma}V'^\sigma \tag{45}$$

Here, to make the derivative of a vector covariant, the following condition must be satisfied:

$$\frac{\partial x^\rho}{\partial x'^\mu}\frac{\partial^2 x'^\nu}{\partial x^\rho \partial x^\lambda}V^\lambda + \Gamma'^\nu{}_{\mu\sigma}V'^\sigma = \frac{\partial x^\rho}{\partial x'^\mu}\frac{\partial x'^\nu}{\partial x^\lambda}\Gamma^\lambda{}_{\rho\sigma}V^\sigma \tag{46}$$

Which is

$$\Gamma'^\nu{}_{\mu\sigma}\left(\frac{\partial x'^\sigma}{\partial x^\tau}V^\tau\right) = \frac{\partial x^\rho}{\partial x'^\mu}\frac{\partial'^\nu}{\partial x^\lambda}\Gamma^\lambda{}_{\rho\sigma}V^\sigma - \frac{\partial x^\rho}{\partial x'^\mu}\frac{\partial x^\rho}{\partial x'^\mu}\frac{\partial^2 x'^\nu}{\partial x^\rho \partial x^\lambda}V^\lambda \tag{47}$$

$$\Gamma'^\nu{}_{\mu\kappa}V^\tau = \frac{\partial x^\rho}{\partial x'^\kappa}\frac{\partial x^\rho}{\partial x'^\mu}\frac{\partial x'^\nu}{\partial x^\lambda}\Gamma^\lambda{}_{\rho\sigma}V^\sigma - \frac{\partial x^\tau}{\partial x'^\kappa}\frac{\partial x^\rho}{\partial x'^\mu}\frac{\partial^2 x'^\nu}{\partial x^\rho \partial x^\lambda}V^\lambda \tag{48}$$

This leads to the explicit form of how the Christoffel symbols transform under coordinate changes.

$$\Gamma'^\nu{}_{\mu\kappa} = \frac{\partial x^\tau}{\partial x'^\kappa}\frac{\partial x^\rho}{\partial x'^\mu}\frac{\partial x'^\nu}{\partial x^\lambda}\Gamma^\lambda{}_{\rho\tau} - \frac{\partial x^\tau}{\partial x'^\kappa}\frac{\partial x^\rho}{\partial x'^\mu}\frac{\partial^2 x'^\nu}{\partial x^\rho \partial x^\tau} \tag{49}$$

Since the Kronecker delta is a constant matrix, it is clear that its derivative must vanish. By applying the chain rule to the delta, one can derive the following relation, which simplifies the transformation rule described above.

$$\frac{\partial}{\partial x'^\mu}\delta^\nu_\kappa = \frac{\partial}{\partial x'^\mu}\frac{\partial x'^\nu}{\partial x'^\kappa} = \frac{\partial}{\partial x'^\mu}\left(\frac{\partial x^\tau}{\partial x'^\kappa}\frac{\partial x'^\nu}{\partial x^\tau}\right) = 0 = \frac{\partial x^\tau}{\partial x'^\kappa}\frac{\partial x^\rho}{\partial x'^\mu}\frac{\partial^2 x'^\nu}{\partial x^\rho \partial x^\tau} + \frac{\partial x'^\nu}{\partial x^\tau}\frac{\partial x'^\nu}{\partial x^\tau}\frac{\partial^2 x^\tau}{\partial x'^\mu \partial x'^\rho} \tag{50}$$

Finally, the transformation rule for the Christoffel symbols is given by:

$$\Gamma'^\nu{}_{\mu\kappa} = \frac{\partial x^\tau}{\partial x'^\kappa}\frac{\partial x^\rho}{\partial x'^\mu}\frac{\partial x'^\nu}{\partial x^\lambda}\Gamma^\lambda{}_{\rho\tau} + \frac{\partial x'^\nu}{\partial x^\tau}\frac{\partial^2 x^\tau}{\partial x'^\mu \partial x'^\rho} \tag{51}$$

By the same reasoning, one can easily determine how covariant derivatives act on differential forms.

$$\nabla_\mu V_\nu = \partial_\mu V_\nu - \Gamma^\lambda{}_{\mu\nu}V_\lambda \tag{52}$$

### B.5 EXPLICIT FORM OF CHRISTOFFEL SYMBOL

The metric serves as the ruler of a given geometry; therefore, it should remain invariant with respect to position in a coordinate system. In the case of Euclidean space, this invariance is trivial to observe, as the metric is simply $\delta_{\mu\nu}$, a constant matrix.

$$\frac{\partial}{\partial x^\lambda} \delta_{\mu\nu} = 0 \tag{53}$$

However, in the curved case, the above principle must still hold to interpret the metric as a ruler. Nevertheless, this condition does not hold when using an ordinary derivative. Here, the covariant derivative comes into play, replacing the ordinary derivative. When taking the covariant derivative of the curved metric, the resulting term vanishes.

$$\nabla_\lambda g_{\mu\nu} = 0 \tag{54}$$

One can express this condition in terms of a flat metric combined with a diffeomorphism transformation factor.

$$g_{\mu\nu}(x) = \frac{\partial \xi^\lambda}{\partial x^\mu} \frac{\partial \xi^\rho}{\partial x^\nu} \delta_{\lambda\rho}(\xi) \tag{55}$$

Taking the derivative with respect to $x$ on both sides, the equation becomes:

$$\frac{\partial}{\partial x^\sigma} g_{\mu\nu}(x) = \frac{\partial^2 x^\lambda}{\partial x^\sigma \partial x^\mu} \frac{\xi^\rho}{\partial x^\nu} \delta_{\lambda\rho} + \frac{\partial^2 \xi^\rho}{\partial x^\sigma \partial x^\nu} \frac{\partial \xi^\lambda}{\partial x^\mu} \delta_{\lambda\rho} \tag{56}$$

$$= \frac{\partial^2 \xi^\rho}{\partial x^\sigma \partial x^\nu} \frac{\partial x^\tau}{\partial \xi^\rho} \frac{\partial \xi^\rho}{\partial x^\tau} \frac{\partial \xi^\lambda}{\partial x^\mu} \delta_{\lambda\rho} + \frac{\partial^2 \xi^\lambda}{\partial x^\sigma \partial x^\mu} \frac{\partial x^\tau}{\partial \xi^\lambda} \frac{\partial \xi^\lambda}{\partial x^\tau} \frac{\partial \xi^\rho}{\partial x^\nu} \delta_{\lambda\rho} \tag{57}$$

$$= \frac{\partial^2 \xi^\rho}{\partial x^\sigma \partial x^\nu} \frac{\partial x^\tau}{\partial \xi^\rho} g_{\mu\tau} + \frac{\partial^2 \xi^\lambda}{\partial x^\sigma \partial x^\mu} \frac{\partial x^\tau}{\partial \xi^\lambda} g_{\tau\nu} \tag{58}$$

From Eq. 54, one can easily derive the explicit form of the Christoffel symbol in terms of the derivatives of the curved and flat coordinates.

$$\frac{\partial}{\partial x^\sigma} g_{\mu\nu} = \Gamma^\tau{}_{\sigma\mu} g_{\tau\nu} + \Gamma^\tau{}_{\nu\sigma} g_{\mu\sigma} \tag{59}$$

$$\Gamma^\tau{}_{\sigma\mu} = \frac{\partial^2 \xi^\lambda}{\partial x^\sigma \partial x^\mu} \frac{\partial x^\tau}{\partial \xi^\lambda}(x) \tag{60}$$

Since the metric should always be symmetric, the lower indices of the Christoffel symbol should also be symmetric. It is called a torsion-free condition. Furthermore, by utilizing a simple mathematical trick, one can obtain the Christoffel symbol in terms of the metric $g_{\mu\nu}$.

$$\frac{\partial}{\partial x^\sigma} g_{\mu\nu} = \Gamma^\tau{}_{\sigma\mu} g_{\tau\nu} + \Gamma^\tau{}_{\sigma\nu} g_{\mu\tau} \tag{61}$$

$$\frac{\partial}{\partial x^\mu} g_{\nu\sigma} = \Gamma^\tau{}_{\mu\nu} g_{\tau\sigma} + \Gamma^\tau{}_{\mu\sigma} g_{\nu\tau} \tag{62}$$

$$\frac{\partial}{\partial x^\nu} g_{\sigma\mu} = \Gamma^\tau{}_{\nu\sigma} g_{\tau\mu} + \Gamma^\tau{}_{\nu\mu} g_{\sigma\tau} \tag{63}$$

Adding the first two equations and subtracting the last one leads to:

$$\Gamma^\lambda{}_{\mu\nu} = \frac{1}{2} g^{\lambda\rho} \left( \frac{\partial}{\partial x^\mu} g_{\nu\rho} + \frac{\partial}{\partial x^\nu} g_{\rho\mu} - \frac{\partial}{\partial x^\rho} g_{\mu\nu} \right) \tag{64}$$

### B.6 GEODESIC EQUATIONS

The shortest path between two points is simple to define in flat space. However, in curved space, this notion becomes more complicated. The shortest path in a curved space is defined as a geodesic. There are several ways to derive the geodesic equation, one of which is by imposing the free-fall condition.

$$\frac{\partial^2 \xi^\mu(\tau)}{\partial \tau^2} = 0 \tag{65}$$

By a diffeomorphism, one can transform a coordinate into an arbitrary coordinate $x$.

$$0 = \frac{\partial}{\partial \tau}\left(\frac{\partial \xi^\mu}{\partial x^\nu}\frac{\partial x^\nu}{\partial \tau}\right) = \frac{\partial \xi^\mu}{\partial x^\nu}\frac{\partial^2 x^\nu}{\partial \tau^2} + \frac{\partial^2 \xi^\mu}{\partial x^\lambda \partial x^\nu}\frac{\partial x^\lambda}{\partial \tau}\frac{\partial x^\nu}{\partial \tau} \tag{66}$$

$$\frac{\partial^2 x^\rho}{\partial \tau^2} + \frac{\partial^2 \xi^\mu}{\partial x^\lambda \partial x^\nu}\frac{\partial x^\rho}{\partial \xi^\mu}\frac{\partial x^\lambda}{\partial \tau}\frac{\partial x^\nu}{\partial \tau} = \frac{\partial^2 x^\rho}{\partial \tau^2} + \Gamma^\rho{}_{\lambda\nu}\frac{\partial x^\lambda}{\partial \tau}\frac{\partial x^\nu}{\partial \tau} = 0 \tag{67}$$

Another way to derive the equation is by minimizing the distance in curved space.

$$S = \int \sqrt{g_{\mu\nu}\frac{dx^\mu}{d\tau}\frac{dx^\nu}{d\tau}}d\tau \tag{68}$$

By varying the above equation and requiring the variation to vanish, one can compute its minimum value, and after some tedious calculations, the geodesic equation can be obtained.

### B.7 RIEMANN CURVATURE

The Riemann curvature can be defined through the concept of parallel transport of a vector. In flat space, a vector remains unchanged under parallel transport along any path. However, in curved space, the vector's outcome depends on the path taken. This leads to the idea of curvature as the difference between the results of transporting a vector along two different paths from the same starting point to the same endpoint. This difference quantitatively characterizes the curvature of the space.

$$[\nabla_\mu, \nabla_\nu]V^\lambda \tag{69}$$

Here, the bracket denotes the commutation relation between the entities. Since the covariant derivative acts as the generator of parallel transport, the equation can be interpreted as the vector $V^\lambda$ being transported along two different paths: one generated by applying $\nabla_\mu$ followed by $\nabla_\nu$, and the other by reversing the order. The resulting computation takes the form:

$$\begin{aligned}
\nabla_\mu \nabla_\nu V^\lambda &= \partial_\mu(\nabla_\nu V^\lambda) + \Gamma^\lambda{}_{\mu\rho}\nabla_\nu V^\rho - \Gamma^\rho{}_{\mu\nu}\nabla_\rho V^\lambda \\
&= \partial_\mu \partial_\nu V^\lambda + \partial_\mu \Gamma^\lambda{}_{\nu\rho}V^\rho + \Gamma^\lambda{}_{\nu\rho}\partial_\mu V^\rho + \Gamma^\lambda{}_{\mu\rho}\partial_\nu V^\rho \\
&\quad + \Gamma^\lambda{}_{\mu\rho}\Gamma^\rho{}_{\nu\sigma}V^\sigma - \Gamma^\rho{}_{\mu\nu}\partial_\rho V^\lambda - \Gamma^\rho{}_{\mu\nu}\Gamma^\lambda{}_{\rho\sigma}V^\sigma
\end{aligned} \tag{70}$$

Where,

$$[\nabla_\mu, \nabla_\nu]V^\lambda = (\partial_\mu \Gamma^\lambda{}_{\nu\rho} - \partial_\nu \Gamma^\lambda{}_{\mu\rho} + \Gamma^\lambda{}_{\mu\sigma}\Gamma^\sigma{}_{\nu\rho} - \Gamma^\lambda{}_{\nu\sigma}\Gamma^\sigma{}_{\mu\rho})V^\rho - 2\Gamma^\rho{}_{[\mu\nu]}\nabla_\rho V^\lambda \tag{71}$$

Since the connection is symmetric under the permutation of its lower indices, the last term in the above equation can be eliminated. We can then finally define the Riemann tensor.

$$R^\lambda{}_{\rho\mu\nu} := \partial_\mu \Gamma^\lambda{}_{\nu\rho} - \partial_\nu \Gamma^\lambda{}_{\mu\rho} + \Gamma^\lambda{}_{\mu\sigma}\Gamma^\sigma{}_{\nu\rho} - \Gamma^\lambda{}_{\nu\sigma}\Gamma^\sigma{}_{\mu\rho} \tag{72}$$

The Riemann curvature tensor possesses several useful properties.

$$\begin{aligned}
R_{\lambda\rho\mu\nu} &= -R_{\lambda\rho\nu\mu} \\
R_{\lambda\rho\mu\nu} &= -R_{\rho\lambda\nu\mu} \\
R_{\lambda\rho\mu\nu} &= -R_{\nu\mu\lambda\rho} \\
R_{\lambda\rho\mu\nu} + R_{\lambda\mu\nu\rho} + R_{\lambda\nu\rho\mu} &= 0 \\
\nabla_\sigma R_{\lambda\rho\mu\nu} + \nabla_\mu R_{\lambda\rho\nu\sigma} + \nabla_\nu R_{\lambda\rho\sigma\mu} &= 0 \\
\nabla_\sigma R_{\lambda\rho\mu\nu} + \nabla_\lambda R_{\rho\sigma\nu\mu} + \nabla_\rho R_{\sigma\lambda\mu\nu} &= 0 \\
R_{\lambda\rho\mu\nu} + R_{\rho\mu\lambda\nu} + R_{\mu\lambda\rho\nu} &= 0
\end{aligned} \tag{73}$$

With the curved metric $g_{\mu\nu}$, one can construct the Ricci curvature tensor and the Ricci scalar by contracting the first and third, and the second and fourth indices of the Riemann tensor, respectively.

$$\begin{aligned}
R_{\rho\nu} &= g^{\lambda\mu}R_{\lambda\rho\mu\nu} \\
R &= g^{\rho\nu}R_{\rho\nu} = g^{\rho\nu}g^{\lambda\mu}R_{\lambda\rho\mu\nu}
\end{aligned} \tag{74}$$

## C CURVATURE COMPUTATION FOR TWO LAYERED MLP

Before we proceed, it is convenient to define some symbols that will frequently appear in the following calculations. Since we will use the SiLU activation function, the logistic function will

appear repeatedly in derivative computations. Therefore, we introduce the following symbol to represent the logistic function:

$$\text{LS}(x) \equiv \frac{1}{1 + e^{-x}} \tag{75}$$

The derivative of the logistic function is well known. It consists of the square of the logistic function multiplied by an $xe^{-x}$ term.

$$\frac{d}{dx}\text{LS}(x) = e^{-x} \times \text{LS}(x)^2 = \frac{e^{-x}}{(1 + e^{-x})^2} \tag{76}$$

The coordinate transformation function in the model is based on an MLP with the SiLU activation function. To obtain the Jacobian of the transformation function, one must differentiate the transformation function with respect to the transformed coordinate $x'$.

$$J^i_{\ j} = \frac{dx'^i}{dx^j} \tag{77}$$

Here, $x'^i$ can be expressed in the following form.

$$x'^i = W^{(2)i}_{\ \ k} f(W^{(1)k}_{\ \ j} x^j + b^{(1)k}) + b^{(2)i} \tag{78}$$

$W^{(n)i}_{\ \ j}$, $b^{(n)i}$, and $f(x)$ denote the weight matrix, bias for each distinct hidden layer, and activation function, respectively. The activation function is, in this case, SiLU. Thus, the derivative of the function becomes straightforward.

$$\frac{dx'^i}{dx^j} = W^{(2)i}_{\ \ m} W^{(1)m}_{\ \ k} f(W^k_{\ l} x^l + b^k)_{,j} \tag{79}$$

The derivative of the activation term is tedious but manageable. First, we will show how the derivative of the SiLU function appears.

$$\text{SiLU}(x) \equiv x \times \text{LS}(x) \tag{80}$$

Hence, the derivative can be expressed as follows:

$$\frac{d}{dx}\text{Silu}(x) = x \times \text{LS}(x)' + \text{LS}(x) = \frac{xe^{-x}}{(1 + e^{-x})^2} + \frac{1}{1 + e^{-x}} \tag{81}$$

For computational convenience, we first derive the derivative of the exponential term with respect to the weight and bias.

$$\partial_k e^{-(W^i_{\ j} x^j + b^i)} = -W^l_{\ k}(e^{W^l_{\ j} x^j + b^l})^i_{\ l} = -W^l_{\ k} E^i_{\ l} \tag{82}$$

Where $(e^{-(W^l_{\ j} x^j + b^l)})^i_{\ l}$ is a diagonal form as follows.

$$\text{E}^i_{\ l} \equiv (e^{-(W^l_{\ j} x^j + b^j)})^i_{\ l} = \begin{cases} e^{-(W^l_{\ j} x^j + b^l)} & \text{if } l = i \\ 0 & \text{if } l \neq i \end{cases} \tag{83}$$

By plugging Eq.81 into Eq.79, one can obtain the final form of the Jacobian. To express the equation in a simpler form, it is convenient to introduce the following symbols before proceeding with the main computation.

$$\sigma^i = \frac{1}{1 + e^{-(W^{(1)i}_{\ \ j} x^j + b^{(1)i})}} \tag{84}$$

$$\partial_j \sigma^i = W^{(1)l}_{\ \ j} E^i_{\ l}(\sigma^2)^i \tag{85}$$

$$x'^i = W^{(1)i}_{\ \ j} x^j + b^{(1)i} = (W^{(1)}x + b^{(1)})^i \tag{86}$$

Then, the Jacobian can be expressed in terms of the symbols introduced above.

$$\begin{aligned} \frac{dx'^i}{dx^j} &= W^{(2)i}_{\ \ m} W^{(1)m}_{\ \ k}(((x^{(1)k})e^{-x^{(1)k}} \times \text{LS}(x^{(1)k}) + 1)\text{LS}(x^{(1)k}))^k_{\ j} \\ &= \sum_{a_1} W^{(2)i}_{\ \ a_1}(W^{(1)a_1}_{\ \ j}\sigma^{a_1} + (W^{(1)}x + b^{(1)})^{a_1} W^{(1)a_3}_{\ \ j} E^{a_1}_{\ a_3}(\sigma^2)^{a_1}) \end{aligned} \tag{87}$$

Due to the diffeomorphism invariance of a Riemannian manifold, the metric tensor can be decomposed into the square of the Jacobian of the given coordinate transformation, coupled with vectors and the flat Euclidean metric.

$$g_{ij} = \frac{dx'^m}{dx^i}\frac{dx'^n}{dx^j}\eta_{mn} = \frac{dx'^m}{dx^i}\frac{dx'_m}{dx^j} \tag{88}$$

where the metric can be explicitly expressed using Eq. 79.

$$g_{ij} = \sum_{a_1 a_6} W^{(2)a_4}{}_{a_1}(W^{(1)a_1}{}_i\sigma^{a_1} + (W^{(1)}x + b^{(1)})^{a_1}W^{(1)a_3}{}_iE^{a_1}{}_{a_3}(\sigma^2)^{a_1}) \\ W^{(2)}{}_{a_4 a_6}(W^{(1)a_6}{}_j\sigma^{a_6} + (W^{(1)}x + b^{(1)})^{a_6}W^{(1)a_7}{}_jE^{a_6}{}_{a_7}(\sigma^2)^{a_6}) \tag{89}$$

As shown above, the curved metric $g_{ij}$ can be expressed in terms of the weight, bias, and input vector $x$. By taking derivatives and appropriately contracting the metric, the curvature tensor can also be expressed in terms of these components. The curvature tensor consists of combinations of derivatives of the Christoffel symbols. To compute a Christoffel symbol, one must first obtain the derivative of the metric tensor. The derivative of the metric tensor can be expressed as follows:

$$\partial_k g_{ij} = \left(\frac{\partial^2 x'^m}{\partial x^k \partial x^i}\frac{\partial x'^n}{\partial x^j} + \frac{\partial^2 x'^n}{\partial x^k \partial x^j}\frac{\partial x'^m}{\partial x^i}\right)\eta_{mn} \tag{90}$$

Here, the key term is the second derivative of a vector. To compute this second derivative, one must consider the derivative of the $E^{ij}$ term.

$$\partial_k E^i{}_j = -W^l{}_k\delta^i{}_{lp}E^p{}_j \tag{91}$$

By utilizing the relation above, one can compute the second derivative of an arbitrary vector $x'^i$, which is a crucial component for deriving the affine connection.

$$\frac{\partial^2 x'^i}{\partial x^k \partial x^j} = \sum_{a_1} W^{(2)i}{}_{a_1}(W^{(1)a_1}{}_jW^{(1)a_2}{}_kE^{a_1}{}_{a_2}(\sigma^2)^{a_1} + W^{(1)a_1}{}_kW^{(1)a_3}{}_jE^{a_1}{}_{a_3}(\sigma^2)^{a_1} \\ -(W^{(1)}x + b^{(1)})^{a_1}W^{(1)a_3}{}_jW^{(1)a_4}{}_k\delta^{a_1}{}_{a_4 a_5}E^{a_5}{}_{a_3}(\sigma^2)^{a_1} \\ +2(W^{(1)}x + b^{(1)})^{a_1}W^{(1)a_3}{}_jE^{a_1}{}_{a_3}W^{(1)a_6}{}_kE^{a_1}{}_{a_6}(\sigma^3)^{a_1}) \tag{92}$$

Now, with the second derivative, we can construct the derivative of the metric.

$$\partial_k g_{ij} = \sum_{a_1 a_7}(W^{(2)m}{}_{a_1}(W^{(1)a_1}{}_iW^{(1)a_2}{}_kE^{a_1}{}_{a_2}(\sigma^2)^{a_1} + W^{(1)a_1}{}_kW^{(1)a_3}{}_iE^{a_1}{}_{a_3}(\sigma^2)^{a_1} \\ -(W^{(1)}x + b^{(1)})^{a_1}W^{(1)a_3}{}_iW^{(1)a_4}{}_k\delta^{a_1}{}_{a_4 a_5}E^{a_5}{}_{a_3}(\sigma^2)^{a_1} \\ +2(W^{(1)}x + b^{(1)})^{a_1}W^{(1)a_3}{}_iE^{a_1}{}_{a_3}W^{(1)a_6}{}_kE^{a_1}{}_{a_6}(\sigma^3)^{a_1}) \\ W^{(2)n}{}_{a_7}(W^{(1)a_7}{}_j\sigma^{a_7} + (W^{(1)}x + b^{(1)})^{a_7}W^{(1)a_8}{}_jE^{a_7}{}_{a_8}(\sigma^2)^{a_7}) \\ +W^{(2)n}{}_{a_1}(W^{(1)a_1}{}_jW^{(1)a_2}{}_kE^{a_1}{}_{a_2}(\sigma^2)^{a_1} + W^{(1)a_1}{}_kW^{(1)a_3}{}_jE^{a_1}{}_{a_3}(\sigma^2)^{a_1} \\ -(W^{(1)}x + b^{(1)})^{a_1}W^{(1)a_3}{}_jW^{(1)a_4}{}_k\delta^{a_1}{}_{a_4 a_5}E^{a_5}{}_{a_3}(\sigma^2)^{a_1} \\ +2(W^{(1)}x + b^{(1)})^{a_1}W^{(1)a_3}{}_jE^{a_1}{}_{a_3}W^{(1)a_6}{}_kE^{a_1}{}_{a_6}(\sigma^3)^{a_1}) \\ W^{(2)m}{}_{a_7}(W^{(1)a_7}{}_i\sigma^{a_7} + (W^{(1)}x + b^{(1)})^{a_7}W^{(1)a_8}{}_iE^{a_7}{}_{a_8}(\sigma^2)^{a_7}))\eta_{mn} \tag{93}$$

Since the Christoffel symbol can be written in terms of the derivative of the given metric, one can now express the complete form of the symbol using Eq. 93.

$$\Gamma^i{}_{jk} = \frac{1}{2}g^{im}(\partial_j g_{mk} + \partial_k g_{mj} - \partial_m g_{kj}) \tag{94}$$

Here, $g^{im}$ is the inverse of the metric tensor, which satisfies the following relation:

$$g^{ij}g_{jk} = \delta^i{}_k \tag{95}$$
$$g^{ij}g_{ji} = D \tag{96}$$

where $D$ is the number of spatial dimensions. The inverse of the metric can be explicitly written using the inverse Jacobian.

$$g^{ij} = \frac{dx^i}{dx'^m}\frac{dx^j}{dx'^n}\eta^{mn} \tag{97}$$

Although the explicit form of the inverse metric is written in terms of combinations of inverse Jacobians, we design two distinct models to encapsulate the Jacobian and the inverse Jacobian

separately for each case. Thus, the inverse Jacobian is not the actual matrix inverse of the Jacobian, but instead follows the same computational process as the Jacobian, with the model replaced by the inverse transfer model. Using this setup, one can then express the explicit form of the inverse metric tensor in terms of weights and biases.

$$
g^{ij} = \sum_{a_1 a_6} W'^{(2)a_4}{}_{a_1} (W'^{(1)a_1 i}\sigma^{a_1} + (W'^{(1)}x + b^{(1)})^{a_1} W'^{(1)a_3 i} E^{a_1}{}_{a_3}(\sigma^2)^{a_1})
$$
$$
W'^{(2)}{}_{a_4 a_6} (W'^{(1)a_6 j}\sigma^{a_6} + (W'^{(1)}x + b^{(1)})^{a_6} W'^{(1)a_7 j} E^{a_6}{}_{a_7}(\sigma^2)^{a_6}) \tag{98}
$$

The primed weights and biases indicate the weights and biases from the inverse transfer model. Finally, all individual components are now prepared to complete the expression for the Christoffel symbol. We now recall Eq. 94.

$$
\begin{aligned}
\Gamma^i{}_{jk} =\ & \tfrac{1}{2} g^{im}(\partial_j g_{mk} + \partial_k g_{mj} - \partial_m g_{kj}) \\
=\ & \tfrac{1}{2} \sum_{a_1 a_4} W'^{(2)a_3}{}_{a_1} (W'^{(1)a_1 i}\sigma^{a_1} + (W'^{(1)}x + b^{(1)})^{a_1} W'^{(1)a_2 i} E^{a_1}{}_{a_2}(\sigma^2)^{a_1}) \\
& W'^{(2)}{}_{a_3 a_4} (W'^{(1)a_4 m}\sigma^{a_4} + (W'^{(1)}x + b^{(1)})^{a_4} W'^{(1)a_5 m} E^{a_4}{}_{a_5}(\sigma^2)^{a_4}) \\
& \Big( \sum_{a_5 a_{11}} W^{(2)o}{}_{a_5} ((W^{(1)a_5}{}_m W^{(1)a_6}{}_j E^{a_5}{}_{a_6}(\sigma^2)^{a_5} + W^{(1)a_5}{}_j W^{(1)a_7}{}_m E^{a_5}{}_{a_7}(\sigma^2)^{a_5} \\
& - (W^{(1)}x + b^{(1)})^{a_5} W^{(1)a_7}{}_m W^{(1)a_8}{}_j \delta^{a_5}{}_{a_8 a_9} E^{a_9}{}_{a_7}(\sigma^2)^{a_5} \\
& + 2(W^{(1)}x + b^{(1)})^{a_5} W^{(1)a_7}{}_m E^{a_5}{}_{a_7} W^{(1)a_{10}}{}_j E^{a_5}{}_{a_{10}}(\sigma^3)^{a_5}) \\
& W^{(2)p}{}_{a_{11}} (W^{(1)a_{11}}{}_k \sigma^{a_{11}} + (W^{(1)}x + b^{(1)})^{a_{11}} W^{(1)a_{12}}{}_k E^{a_{11}}{}_{a_{12}}(\sigma^2)^{a_{11}}) \\
& + W^{(2)p}{}_{a_5} (W^{(1)a_5}{}_k W^{(1)a_6}{}_j E^{a_5}{}_{a_6}(\sigma^2)^{a_5} + W^{(1)a_5}{}_j W^{(1)a_7}{}_k E^{a_5}{}_{a_7}(\sigma^2)^{a_5} \\
& - (W^{(1)}x + b^{(1)})^{a_5} W^{(1)a_7}{}_k W^{(1)a_8}{}_j \delta^{a_5}{}_{a_8 a_9} E^{a_9}{}_{a_7}(\sigma^2)^{a_5} \\
& + 2(W^{(1)}x + b^{(1)})^{a_5} W^{(1)a_7}{}_k E^{a_5}{}_{a_7} W^{(1)a_{10}}{}_j E^{a_5}{}_{a_{10}}(\sigma^3)^{a_5}) \\
& W^{(2)o}{}_{a_{11}} (W^{(1)a_{11}}{}_m \sigma^{a_{11}} + (W^{(1)}x + b^{(1)})^{a_{11}} W^{(1)a_{12}}{}_m E^{a_{11}}{}_{a_{12}}(\sigma^2)^{a_{11}}))\eta_{op} \\
& + W^{(2)o}{}_{a_5} ((W^{(1)a_5}{}_m W^{(1)a_6}{}_k E^{a_5}{}_{a_6}(\sigma^2)^{a_5} + W^{(1)a_5}{}_k W^{(1)a_7}{}_m E^{a_5}{}_{a_7}(\sigma^2)^{a_5} \\
& - (W^{(1)}x + b^{(1)})^{a_5} W^{(1)a_7}{}_m W^{(1)a_8}{}_k \delta^{a_5}{}_{a_8 a_9} E^{a_9}{}_{a_7}(\sigma^2)^{a_5} \\
& + 2(W^{(1)}x + b^{(1)})^{a_5} W^{(1)a_7}{}_m E^{a_5}{}_{a_7} W^{(1)a_{10}}{}_k E^{a_5}{}_{a_{10}}(\sigma^3)^{a_5}) \\
& W^{(2)p}{}_{a_{11}} (W^{(1)a_{11}}{}_j \sigma^{a_{11}} + (W^{(1)}x + b^{(1)})^{a_{11}} W^{(1)a_{12}}{}_j E^{a_{11}}{}_{a_{12}}(\sigma^2)^{a_{11}}) \\
& + W^{(2)p}{}_{a_5} (W^{(1)a_5}{}_j W^{(1)a_6}{}_k E^{a_5}{}_{a_6}(\sigma^2)^{a_5} + W^{(1)a_5}{}_k W^{(1)a_7}{}_j E^{a_5}{}_{a_7}(\sigma^2)^{a_5} \\
& - (W^{(1)}x + b^{(1)})^{a_5} W^{(1)a_7}{}_j W^{(1)a_8}{}_k \delta^{a_5}{}_{a_8 a_9} E^{a_9}{}_{a_7}(\sigma^2)^{a_5} \\
& + 2(W^{(1)}x + b^{(1)})^{a_5} W^{(1)a_7}{}_j E^{a_5}{}_{a_7} W^{(1)a_{10}}{}_k E^{a_5}{}_{a_{10}}(\sigma^3)^{a_5}) \\
& W^{(2)o}{}_{a_{11}} (W^{(1)a_{11}}{}_m \sigma^{a_{11}} + (W^{(1)}x + b^{(1)})^{a_{11}} W^{(1)a_{12}}{}_m E^{a_{11}}{}_{a_{12}}(\sigma^2)^{a_{11}}))\eta_{op} \\
& - W^{(2)o}{}_{a_5} ((W^{(1)a_5}{}_k W^{(1)a_6}{}_m E^{a_5}{}_{a_6}(\sigma^2)^{a_5} + W^{(1)a_5}{}_m W^{(1)a_7}{}_k E^{a_5}{}_{a_7}(\sigma^2)^{a_5} \\
& - (W^{(1)}x + b^{(1)})^{a_5} W^{(1)a_7}{}_k W^{(1)a_8}{}_m \delta^{a_5}{}_{a_8 a_9} E^{a_9}{}_{a_7}(\sigma^2)^{a_5} \\
& + 2(W^{(1)}x + b^{(1)})^{a_5} W^{(1)a_7}{}_k E^{a_5}{}_{a_7} W^{(1)a_{10}}{}_m E^{a_5}{}_{a_{10}}(\sigma^3)^{a_5}) \\
& W^{(2)p}{}_{a_{11}} (W^{(1)a_{11}}{}_j \sigma^{a_{11}} + (W^{(1)}x + b^{(1)})^{a_{11}} W^{(1)a_{12}}{}_j E^{a_{11}}{}_{a_{12}}(\sigma^2)^{a_{11}}) \\
& + W^{(2)p}{}_{a_5} (W^{(1)a_5}{}_j W^{(1)a_6}{}_m E^{a_5}{}_{a_6}(\sigma^2)^{a_5} + W^{(1)a_5}{}_m W^{(1)a_7}{}_j E^{a_5}{}_{a_7}(\sigma^2)^{a_5} \\
& - (W^{(1)}x + b^{(1)})^{a_5} W^{(1)a_7}{}_j W^{(1)a_8}{}_m \delta^{a_5}{}_{a_8 a_9} E^{a_9}{}_{a_7}(\sigma^2)^{a_5} \\
& + 2(W^{(1)}x + b^{(1)})^{a_5} W^{(1)a_7}{}_j E^{a_5}{}_{a_7} W^{(1)a_{10}}{}_m E^{a_5}{}_{a_{10}}(\sigma^3)^{a_5}) \\
& W^{(2)o}{}_{a_{11}} (W^{(1)a_{11}}{}_k \sigma^{a_{11}} + (W^{(1)}x + b^{(1)})^{a_{11}} W^{(1)a_{12}}{}_k E^{a_{11}}{}_{a_{12}}(\sigma^2)^{a_{11}}))\eta_{op} \Big)
\end{aligned} \tag{99}
$$

To compute the Riemann curvature of the given manifold, one must calculate the second derivative of the metric, as required by its definition.

$$
R^i{}_{ljk} T^l = [\nabla_j, \nabla_k] T^i \tag{100}
$$

where $\nabla_j$ is the covariant derivative, which includes the affine connection.

$$
\nabla_j T^i = \partial_j T^i + \Gamma^i{}_{jl} T^l \tag{101}
$$

Furthermore, the bracket indicates the commutation relation between the elements; hence, the curvature can be expressed in the following way:

$$\begin{aligned}
R^i{}_{ljk}T^l &= \nabla_j\nabla_k T^i - \nabla_k\nabla_j T^i \\
&= (\partial_j\Gamma^i{}_{kl} - \partial_k\Gamma^i{}_{jl} + \Gamma^i{}_{jm}\Gamma^m{}_{kl} - \Gamma^i{}_{km}\Gamma^m{}_{jl})T^l
\end{aligned} \tag{102}$$

Here, the derivative of the affine connection consists of combinations of second derivatives of the metric tensor.

$$\begin{aligned}
\partial_j\Gamma^i{}_{kl} &= \tfrac{1}{2}\partial_j(g^{im}(\partial_k g_{ml} + \partial_l g_{mk} - \partial_m g_{lk})) \\
&= \tfrac{1}{2}(\partial_j g^{im}(\partial_k g_{ml} + \partial_l g_{mk} - \partial_m g_{lk}) \\
&\quad + g^{im}(\partial_j\partial_k g_{ml} + \partial_j\partial_l g_{mk} - \partial_j\partial_m g_{lk}))
\end{aligned} \tag{103}$$

Therefore, by obtaining the specific form of the second derivative of the metric, one can express the explicit form of the Riemann curvature. To begin the computation, it is convenient to recall Eq.93 for taking the derivative, as well as Eq.82 and Eq. 91 for computing the elements involving the activation function.

$$\begin{aligned}
\partial_j\partial_k g_{ml} &= \sum_{a_1 a_7}\big(W^{(2)o}{}_{a_1}(-2W^{(1)a_1}{}_m W^{(1)a_2}{}_k W^{(1)a_9}{}_j\delta^{a_1}{}_{a_9 a_{10}}E^{a_{10}}{}_{a_2}(\sigma^2)^{a_1} \\
&\quad +4W^{(1)a_1}{}_m W^{(1)a_2}{}_k W^{(1)a_9}{}_j E^{a_1}{}_{a_2}E^{a_1}{}_{a_9}(\sigma^3)^{a_1} \\
&\quad -W^{(1)a_1}{}_j W^{(1)a_3}{}_m W^{(1)a_4}{}_k\delta^{a_1}{}_{a_4 a_5}E^{a_5}{}_{a_3}(\sigma^2)^{a_1} \\
&\quad +(W^{(1)}x+b^{(1)})^{a_1}W^{(1)a_3}{}_m W^{(1)a_4}{}_k W^{(1)a_9}{}_j\delta^{a_1}{}_{a_4 a_5}\delta^{a_5}{}_{a_9 a_{10}}E^{a_{10}}{}_{a_3}(\sigma^2)^{a_1} \\
&\quad -2(W^{(1)}x+b^{(1)})^{a_1}W^{(1)a_3}{}_m W^{(1)a_4}{}_k W^{(1)a_9}{}_j\delta^{a_1}{}_{a_4 a_5}E^{a_5}{}_{a_3}E^{a_1}{}_{a_9}(\sigma^3)^{a_1} \\
&\quad +2W^{(1)a_1}{}_j W^{(1)a_3}{}_m W^{(1)a_6}{}_k E^{a_1}{}_{a_3}E^{a_1}{}_{a_6}(\sigma^3)^{a_1} \\
&\quad -4(W^{(1)}x+b^{(1)})^{a_1}W^{(1)a_3}{}_m W^{(1)a_6}{}_k W^{(1)a_9}{}_j\delta^{a_1}{}_{a_9 a_{10}}E^{a_1}{}_{(a_3}E^{a_{10}}{}_{a_6)}(\sigma^3)^{a_1} \\
&\quad +6(W^{(1)}x+b^{(1)})^{a_1}W^{(1)a_3}{}_m W^{(1)a_6}{}_k W^{(1)a_9}{}_j E^{a_1}{}_{a_9}E^{a_1}{}_{a_3}E^{a_1}{}_{a_6}(\sigma^4)^{a_1}) \\
&\quad W^{(2)p}{}_{a_7}(W^{(1)a_7}{}_l\sigma^{a_7} + (W^{(1)}x+b^{(1)})^{a_7}W^{(1)a_8}{}_l E^{a_7}{}_{a_8}(\sigma^2)^{a_7}) \\
&\quad +W^{(2)o}{}_{a_1}(W^{(1)a_1}{}_m W^{(1)a_2}{}_k E^{a_1}{}_{a_2}(\sigma^2)^{a_1} + W^{(1)a_1}{}_k W^{(1)a_3}{}_m E^{a_1}{}_{a_3}(\sigma^2)^{a_1} \\
&\quad -(W^{(1)}x+b^{(1)})^{a_1}W^{(1)a_3}{}_m W^{(1)a_4}{}_k\delta^{a_1}{}_{a_4 a_5}E^{a_5}{}_{a_3}(\sigma^2)^{a_1} \\
&\quad +2(W^{(1)}x+b^{(1)})^{a_1}W^{(1)a_3}{}_m W^{(1)a_6}{}_k E^{a_1}{}_{a_3}E^{a_1}{}_{a_6}(\sigma^3)^{a_1}) \\
&\quad W^{(2)p}{}_{a_7}(W^{(1)a_7}{}_l W^{(1)a_7}{}_{a_{11}}E^{a_{11}}{}_j(\sigma^2)^{a_7} + W^{(1)a_7}{}_j W^{(1)a_8}{}_l E^{a_7}{}_{a_8}(\sigma^2)^{a_7} \\
&\quad -(W^{(1)}x+b^{(1)})^{a_7}W^{(1)a_8}{}_l W^{(1)a_{11}}{}_j\delta^{a_7}{}_{a_{11}a_{12}}E^{a_{12}}{}_{a_8}(\sigma^2)^{a_7} \\
&\quad +2(W^{(1)}x+b^{(1)})^{a_7}W^{(1)a_8}{}_l W^{(1)a_7}{}_{a_{11}}E^{a_7}{}_{a_8}E^{a_{11}}{}_j(\sigma^3)^{a_7}) \\
&\quad +W^{(2)p}{}_{a_1}(-2W^{(1)a_1}{}_{(l}W^{(1)a_2}{}_{k)}W^{(1)a_{11}}{}_j\delta^{a_1}{}_{a_{11}a_{12}}E^{a_{12}}{}_{a_2}(\sigma^2)^{a_1} \\
&\quad +4W^{(1)a_1}{}_{(l}W^{(1)a_2}{}_{k)}W^{(1)a_{11}}{}_j E^{a_1}{}_{a_2}E^{a_1}{}_{a_{11}}(\sigma^3)^{a_1} \\
&\quad -W^{(1)a_1}{}_j W^{(1)a_3}{}_l W^{(1)a_4}{}_k\delta^{a_1}{}_{a_4 a_5}E^{a_5}{}_{a_3}(\sigma^2)^{a_1} \\
&\quad +(W^{(1)}x+b^{(1)})^{a_1}W^{(1)a_3}{}_l W^{(1)a_4}{}_k W^{(1)a_{11}}{}_j\delta^{a_1}{}_{a_4 a_5}\delta^{a_5}{}_{a_{11}a_{12}}E^{a_{12}}_{a_3}(\sigma^2)^{a_1} \\
&\quad -2(W^{(1)}x+b^{(1)})^{a_1}W^{(1)a_3}{}_l W^{(1)a_{11}}{}_j W^{(1)a_4}{}_k\delta^{a_1}{}_{a_4 a_5}E^{a_5}{}_{a_3}E^{a_1}{}_{a_{11}}(\sigma^3)^{a_1} \\
&\quad +2W^{(1)a_1}{}_j W^{(1)a_3}{}_l W^{(1)a_6}{}_k E^{a_1}{}_{a_3}E^{a_1}{}_{a_6}(\sigma^3)^{a_1} \\
&\quad -4(W^{(1)}x+b^{(1)})^{a_1}W^{(1)a_3}{}_l W^{(1)a_6}{}_k W^{(1)a_{11}}{}_j\delta^{a_1}{}_{a_{11}a_{12}}E^{a_{12}}{}_{(a_3}E^{a_1}{}_{a_6)}(\sigma^3)^{a_1} \\
&\quad +6(W^{(1)}x+b^{(1)})^{a_1}W^{(1)a_3}{}_l W^{(1)a_6}{}_k W^{(1)a_{11}}{}_j E^{a_1}{}_{a_3}E^{a_1}{}_{a_6}E^{a_1}{}_{a_{11}}(\sigma^4)^{a_1}) \\
&\quad W^{(2)o}{}_{a_7}(W^{(1)a_7}{}_m\sigma^{a_7} + (W^{(1)}x+b^{(1)})^{a_7}W^{(1)a_8}{}_m E^{a_7}{}_{a_8}(\sigma^2)^{a_7}) \\
&\quad +W^{(2)p}{}_{a_1}(W^{(1)a_1}{}_l W^{(1)a_2}{}_k E^{a_1}{}_{a_2}(\sigma^2)^{a_1} + W^{(1)a_1}{}_l W^{(1)a_3}{}_l E^{a_1}{}_{a_3}(\sigma^2)^{a_1} \\
&\quad -(W^{(1)}x+b^{(1)})^{a_1}W^{(1)a_3}{}_l W^{(1)a_4}{}_k\delta^{a_1}{}_{a_4 a_5}E^{a_5}{}_{a_3}(\sigma^2)^{a_1} \\
&\quad +2(W^{(1)}x+b^{(1)})^{a_1}W^{(1)a_3}{}_l W^{(1)a_6}{}_k E^{a_1}{}_{a_3}E^{a_1}{}_{a_6}(\sigma^3)^{a_1}) \\
&\quad W^{(2)o}{}_{a_7}(2W^{(1)a_7}{}_{(m}W^{(1)a_{11}}{}_{j)}E^{a_7}{}_{a_{11}}(\sigma^2)^{a_7} \\
&\quad -(W^{(1)}x+b^{(1)})^{a_7}W^{(1)a_8}{}_m W^{(1)a_{11}}{}_j\delta^{a_7}{}_{a_{11}a_{12}}E^{a_{12}}{}_{a_8}(\sigma^2)^{a_7} \\
&\quad +2(W^{(1)}x+b^{(1)})^{a_7}W^{(1)a_8}{}_m W^{(1)a_{11}}{}_j E^{a_7}{}_{a_8}(\sigma^3)^{a_7}E^{a_7}{}_{a_{11}})\eta_{op}
\end{aligned} \tag{104}$$

Now, we have collected all the fundamental components needed to compute the Ricci scalar. Although the computation is quite tedious, it can be carried out through brute-force calculation by referring to Eqs. 72, 74, 93, 98, 99, 103, and 104.

## C.1 QUADRATIC CASE FOR COMPUTATION CHECK

This entire sequence is indeed both tedious and complex. Therefore, we introduce the simplest case for each process to verify the validity of the code and the formulas. Here, we set the activation function to the quadratic of the input signal and maintain the number of layers at two. Under these conditions, the transformed vector becomes:

$$x'^i = W^{(2)i}{}_j (W^{(1)}x + b^{(1)})^{2j} + b^{(2)i} \tag{105}$$

Now, the Jacobian can be easily derived from the above equation.

$$J^i{}_j = W^{(2)i}{}_k 2(W^{(1)}x + b^{(1)})^k{}_m W^{(1)m}{}_j \tag{106}$$

Here, $(W^{(1)}x + b^{(1)})^i{}_j$ has a diagonal matrix form as follows:

$$(W^{(1)}x + b^{(1)})^i{}_j = \begin{cases} W^{(1)i}{}_k x^k + b^{(1)i} & \text{if } i = j \\ 0 & \text{if } i \neq j \end{cases} \tag{107}$$

Then, the metric can be written in the following form:

$$g_{ij} = W^{(2)l}{}_k 2(W^{(1)}x + b^{(1)})^k{}_m W^{(1)m}{}_i W^{(2)}{}_{ln} 2(W^{(1)}x + b^{(1)})^n{}_o W^{(1)o}{}_j \tag{108}$$

Finally, one can compute the derivative of the metric.

$$\begin{aligned} \partial_k g_{ij} = \quad & 4|W^{(2)o}{}_q|^2 (W^{(1)})^{2q}{}_p \delta^p{}_{ik} W^{(1)n}{}_o (W^{(1)}x + b^{(1)})^o{}_j \\ & + 4|W^{(2)}{}_{nq}|^2 W^{(1)q}{}_m (W^{(1)}x + b^{(1)})^m{}_i (W^{(1)})^{2n}{}_p \delta^p{}_{jk} \end{aligned} \tag{109}$$

where $|W|^2 = W^T W$ and $W^2 = W^i{}_k W^k{}_j$.

### C.1.1 2-DIM SIMPLEST EXAMPLE FOR SQUARE ACTIVATION

To cross-check the computation results, we hereby introduce the simplest example for metric computation in 2D. The weights and biases for each layer are defined as follows:

$$W^{(1)i}{}_j = \begin{pmatrix} 1 & 2 \\ 3 & 4 \end{pmatrix} \tag{110}$$

$$W^{(2)i}{}_j = \begin{pmatrix} 5 & 6 \\ 7 & 8 \end{pmatrix} \tag{111}$$

$$b^{(1)i} = \begin{pmatrix} 3 \\ 4 \end{pmatrix} \tag{112}$$

$$x = \begin{pmatrix} 1 \\ 2 \end{pmatrix} \tag{113}$$

Then, the Jacobian can be computed as expressed in Eq. 106. We will break down the Jacobian piece by piece and verify the validity of the equation.

$$2(W^{(1)}x + b^{(1)})^m{}_j = \begin{pmatrix} 16 & 0 \\ 0 & 30 \end{pmatrix} \tag{114}$$

By multiplying the weights from both layers, the equation becomes:

$$J^i{}_j = \begin{pmatrix} 620 & 880 \\ 832 & 1184 \end{pmatrix} \tag{115}$$

Finally, the metric can be expressed as the square of the Jacobian.

$$g_{ij} = (J^T)_{ik} J^k{}_j = \begin{pmatrix} 1076624 & 1530688 \\ 1530688 & 2176256 \end{pmatrix} \tag{116}$$

As shown above, the metric is symmetric.

### C.1.2    2-DIM SIMPLEST EXAMPLE FOR SiLU ACTIVATION

In practice, a simple square activation is insufficient to capture the complex structure of curved space. Therefore, we adopt the SiLU activation function to better express the model's geometric structure. The SiLU function behaves similarly to the ReLU activation but enjoys smoothness across the entire domain.

$$\text{SiLU}(x) = x \times \text{LS}(x) = \frac{x}{1 + e^{-x}} \tag{117}$$

Since the SiLU activation contains the inverse of the exponential function in its expression, the input values should be kept smaller than 1 to prevent the activation from converging to a trivial value.

$$W^{(1)i}{}_j = \begin{pmatrix} 0.1 & 0.2 \\ 0.3 & 0.4 \end{pmatrix} \tag{118}$$

$$W^{(2)i}{}_j = \begin{pmatrix} 0.5 & 0.6 \\ 0.7 & 0.8 \end{pmatrix} \tag{119}$$

$$b^{(1)i} = \begin{pmatrix} 0.3 \\ 0.4 \end{pmatrix} \tag{120}$$

$$x = \begin{pmatrix} 0.1 \\ 0.2 \end{pmatrix} \tag{121}$$

Using the input example set described above, one can compute the explicit equations, with the results as follows. We will first introduce the main components used in the calculation. One key component is the sigmoid function, which is utilized in the SiLU computation.

$$\sigma^i = \begin{pmatrix} 0.5866 \\ 0.6248 \end{pmatrix} \tag{122}$$

Another component is the diagonalized exponential term, which appears in the derivative of the vector exponential.

$$E^i{}_j = \begin{pmatrix} 0.7047 & 0 \\ 0 & 0.6005 \end{pmatrix} \tag{123}$$

By combining the two expressions above with the weights and biases, it is possible to obtain the full Jacobian.

$$J^i{}_j = \begin{pmatrix} 0.1676 & 0.2458 \\ 0.2257 & 0.3322 \end{pmatrix} \tag{124}$$

Finally, by squaring the Jacobian, the induced metric $g_{ij}$ can be defined.

$$g_{ij} = (J^T)_{ik} J^k{}_j = \begin{pmatrix} 0.0790 & 0.1161 \\ 0.1161 & 0.1708 \end{pmatrix} \tag{125}$$

As shown above, the metric is well-defined and forms a symmetric structure in this setup as well.

## D    BASE GRAPH NEURAL NETWORK MODEL

In general, molecule is represented in a graph form. Therefore, in order to handle molecule dataset, it is inevitable to utilize graph neural networks. We chose directional message passing network (DMPNN) (Yang et al., 2019) for our backbone, since it outperforms other GNN architectures in molecular domain. Given a graph, DMPNN initializes the hidden state of each edge $(i, j)$ based on its edge feature $E_{ij}$ with node feature $X_i$. At each step $t$, directional edge summarizes incident edges as a message $m_{ij}^{t+1}$ and updates its hidden state to $h_{ij}^{t+1}$.

$$m_{ij}^{t+1} = \sum_{k \in \mathcal{N}(i) \setminus j} h_{ki}^t \tag{126}$$

$$h_{ij}^{t+1} = \text{ReLU}(h_{ij}^0 + W_e m_{ij}^{t+1}) \tag{127}$$

Where $\mathcal{N}(i)$ denotes the set of neighboring nodes and $W_e$ a learnable weight.he hidden states of nodes are updated by aggregating the hidden states of incident edges into message $m_i^{t+1}$, and passing

its concatenation with the node feature $X_i$ into a linear layer followed by ReLU non-linearity

$$m_i^{t+1} = \sum_{j \in \mathcal{N}(i)} h_{ij}^t \tag{128}$$

$$h_i^{t+1} = \text{ReLU}(W_n \text{concat}(X_i, m_i^{t+1})) \tag{129}$$

Similarly, $W_n$ denotes a learnable weight. Assuming DMPNN runs for $T$ timesteps, we use $(X_{out}, E_{out}) = \text{GNN}(A, X, E)$ to denote the output representation matrices containing hidden states of all nodes and edges, respectively (i.e., $X_{out,i} = h_i^T$ and $E_{out,ij} = h_{ij}^T$).

For graph-level prediction, the node representations after the final GNN layer are typically sum-pooled to obtain a single graph representation $h_G = \sum_i h_i$, which is then passed to a FFN prediction layer.

## E    ARCHITECTURE AND HYPERPARAMETERS

The detailed steps of training GEAR are outlined in Algorithm 1. The model architecture consists of five distinct neural networks, with their parameter sizes summarized in Table 1 and 2. As illustrated in Figures 1 and 6, each task comprises an embedding network, encoder network, transfer network, inverse transfer network, and head network.

The embedding network, denoted as $f_m(x)$, adopts the DMPNN (Directed Message Passing Neural Network) architecture with a depth of 3. It converts the input molecular representation $x$ into a latent representation $a$ in the embedding space. The input vector to the embedding module is constructed as follows, using the same featurization scheme as (Yang et al., 2019): atom features are represented using a 134-dimensional one-hot encoded vector that captures atomic properties such as type, degree, formal charge, hybridization, and aromaticity. Bond features are encoded as a 149-dimensional one-hot vector reflecting bond type, conjugation, ring membership, stereochemistry, and atom-pair-derived descriptors.

The encoder network follows a bottleneck architecture implemented as an autoencoder with multilayer perceptrons (MLPs). The output from the encoder, $f_e(a)$, is then passed to both the transfer network and the head network for subsequent processing.

The output of transfer network $f_t(z)$, denoted as $m$, is used to calculate consistency loss. The induced flat metrics $\eta_{(s)ij}$ and $\eta_{(t)ij}$ from the source and target mappings are iterated K times, with $K = 2$ in our setup. $m$ is also fed into inverse transfer network, so that the output from inverse transfer network $f_i(m)$ can be used to calculate autoencoder loss. Both modules are utilized to compute mapping, metric and curvature losses. The output from head network, $f_h \circ f_i(m)$, is used to calculate regression loss and mapping loss. We trained the model for 1000 epochs with batch size 512 while using

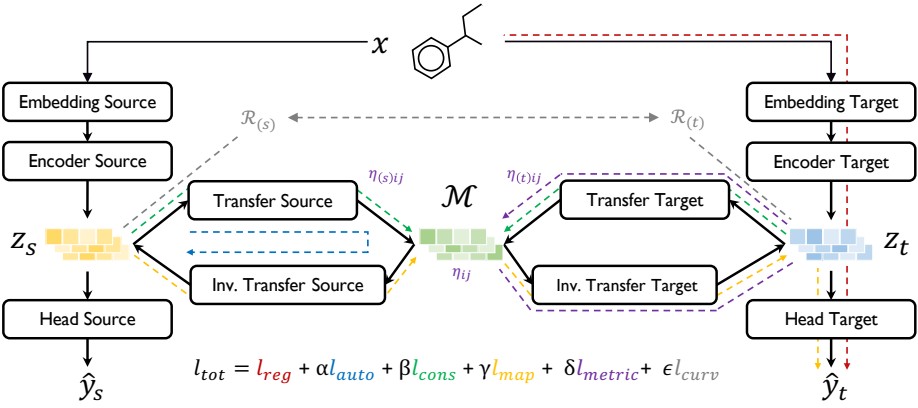

Figure 6: Detailed schematics of GEAR with specific loss function components.

AdamW (Loshchilov & Hutter, 2017) for optimization with learning rate 5e-5. The hyperparameters for $\alpha, \beta, \gamma, \delta, \epsilon$ are 0.1, 0.1, 0.2, 0.1, 0.2 respectively.

Table 1: Common Network Parameters

| network | layer | input, output size | hidden size | dropout |
|---|---|---|---|---|
| embedding | DMPNN | [134, 149], 100 | 200 | 0 |
| encoder | MLP layer | 100, 50 | 50 | 0 |
| transfer | MLP layer | 50, 50 | 50,50,50 | 0.2 |
| inverse transfer | MLP layer | 50, 50 | 50,50,50 | 0.2 |
| head | MLP layer | 50, 1 | 25,12 | 0.2 |

Table 2: Task Specific Encoder Parameters

| Tasks | Random Split Encoder Parameters | Scaffold Split Encoder Parameters |
|---|---|---|
| **hv ← ds** | [200, 200] | [200, 200, 200] |
| **as ← bp** | [200, 200] | [200, 200, 200] |
| **ds ← kri** | [200] | [200, 200] |
| **hv ← vs** | [200, 200] | [200] |
| **vs ← hv** | [200, 200, 200] | [200] |
| **st ← as** | [200, 200, 200] | [200, 200, 200] |
| **ds ← lp** | [200, 200, 200] | [200] |
| **pol ← ds** | [200, 200] | [200, 200, 200] |
| **vs ← bp** | [200, 200, 200] | [200, 200, 200] |
| **dk ← ef** | [200] | [200, 200] |
| **as ← ccs** | [200] | [200, 200, 200] |
| **ct ← bp** | [200, 200] | [200, 200] |
| **st ← ccs** | [200, 200, 200] | [200] |
| **ccs ← kri** | [200] | [200, 200, 200] |
| **hv ← bp** | [200, 200, 200] | [200, 200, 200] |
| **vs ← ccs** | [200, 200, 200] | [200] |
| **st ← hv** | [200, 200] | [200] |
| **hv ← ct** | [200, 200] | [200, 200] |
| **ip ← bp** | [200] | [200] |
| **hv ← ef** | [200, 200] | [200, 200] |
| **hv ← kri** | [100, 100, 100] | [200, 200] |
| **ct ← kri** | [200, 200] | [200, 200, 200] |
| **ip ← dk** | [200] | [200, 200, 200] |

Table 3: Hyperparameters

| | |
|---|---|
| learning rate | 0.00005 |
| optimizer | AdamW |
| batch size | 512 |
| epoch | 1000 |
| $\alpha, \beta, \gamma, \delta, \epsilon$ | 0.1, 0.1, 0.2, 0.1, 0.2 |

---

**Algorithm 1** GEAR

---

1: Initialize embedding network $f_m$, encoder network $f_e$, transfer network $f_t$, inverse transfer network $f_i$, head network $f_h$ with random parameters $\theta$
2: Let $\mathcal{J}(\cdot)$, $\mathcal{G}(\cdot)$, and $\mathcal{R}(\cdot)$ be mathematical functions for jacobian, metric, and curvature computation
3:
4: **for** epoch $i = 1, 2, \ldots n$ **do**
5:     **for** each $t \in Tasks$ **do**
6:         Initialize $L_{metric}, L_{reg}, L_{auto}$ to 0
7:         **for** each batch $\mathbf{b} = (x^t, y^t) \in$ dataset $D$ **do**
8:             $a^t \leftarrow f_m^t(x^t)$
9:             $z^t \leftarrow f_e^t(a^t)$
10:           $m^t \leftarrow f_t^t(z^t)$
11:           $g_{curved}^t \leftarrow \mathcal{G}(\mathcal{J}(m^t, f_i^t))$
12:           $g_{flat}^t \leftarrow \mathcal{G}(\mathcal{J}(f_i^t(m^t, f_t^t), g_{curved}^t))$
13:           $r^t \leftarrow \mathcal{R}(g_{curved}^t)$
14:           **for** step $k = 1, \ldots, K$ **do**
15:             $g_{\text{curved}}^{t,(k)} \leftarrow \mathcal{G}(\mathcal{J}(m^t, f_i^t), g_{\text{flat}}^{t,(k-1)})$
16:             $g_{\text{flat}}^{t,(k)} \leftarrow \mathcal{G}(\mathcal{J}(f_i^t(m^t, f_t^t)), g_{\text{curved}}^{t,(k-1)})$
17:           **end for**
18:           $L_{metric} \leftarrow \sum_{k=1}^{K} \left( \text{MSE Loss}(g_{\text{flat}}^{t,(k)}, I) + \text{MSE Loss}(g_{\text{curved}}^{t,(k)}, I) \right)$
19:           $L_{reg} \leftarrow \text{MSE Loss}(y^t, f_h^t(z^t))$
20:           $L_{auto} \leftarrow \text{MSE Loss}(f_i^t(m^t), z^t)$
21:
22:           **for** each $s \in Subtasks$ **do**
23:             Initialize $L_{map}, L_{cons}, L_{curv}$ to 0
24:             $z^s \leftarrow f_e^s(a^t)$
25:             $m^s \leftarrow f_t^s(z^s)$
26:             $g_{curved}^s \leftarrow \mathcal{G}(\mathcal{J}(m^s, f_i^s))$
27:             $g_{flat}^s \leftarrow \mathcal{G}(\mathcal{J}(f_i^s(m^s, f_s^t), g_{curved}^s))$
28:             $r^s \leftarrow \mathcal{R}(g_{curved}^s)$
29:             **for** step $k = 1, \ldots, K$ **do**
30:               $g_{\text{curved}}^{s,(k)} \leftarrow \mathcal{G}(\mathcal{J}(m^s, f_i^s), g_{\text{flat}}^{s,(k-1)})$
31:               $g_{\text{flat}}^{s,(k)} \leftarrow \mathcal{G}(\mathcal{J}(f_i^s(m^s, f_t^s)), g_{\text{curved}}^{s,(k-1)})$
32:             **end for**
33:             $L_{metric} \leftarrow L_{metric} + \sum_{k=1}^{K} \left( \text{MSE Loss}(g_{\text{flat}}^{s,(k)}, I) + \text{MSE Loss}(g_{\text{curved}}^{s,(k)}, I) \right)$
34:             $L_{map} \leftarrow L_{map} + \text{MSE Loss}(y^t, f_h^t \circ f_i^t(m^s))$
35:             $L_{cons} \leftarrow L_{cons} + \text{MSE Loss}(m^t, m^s)$
36:             $L_{curv} \leftarrow L_{curv} + \text{MSE Loss}(r^t, r^s)$
37:           **end for**
38:
39:           **Compute** $L_{total} = L_{reg} + \alpha L_{auto} + \beta L_{map} + \gamma L_{cons} + \delta L_{metric} + \epsilon L_{curv}$
40:           Update $\theta$ using $L_{total}$
41:         **end for**
42:     **end for**
43: **end for**

---

## F DETAILED EXPLANATION OF DATASETS AND EXPERIMENTAL SETUPS

### F.1 DATASETS

We utilized 14 different molecular property datasets sourced from three open-access databases, as detailed in Table 4 and the descriptions below, for the evaluation of GEAR. Prior to training, the datasets were carefully curated to remove entries with incorrectly specified units, typographical errors,

Table 4: Detailed information about the datasets.

| name | acronym | source | count | mean | std |
|------|---------|--------|-------|------|-----|
| Abraham Descriptor S | AS | Ochem | 1925 | 1.05 | 0.68 |
| Boiling Point | BP | Pubchem | 7139 | 198.99 | 108.88 |
| Collision Cross Section | CCS | Pubchem | 4006 | 205.06 | 57.84 |
| Critical Temperature | CT | Ochem | 242 | 626.04 | 120.96 |
| Dielectric Constant | DK | Ochem | 1007 | 0.80 | 0.41 |
| Density | DS | Pubchem | 3079 | 1.07 | 0.29 |
| Enthalpy of Fusion | EF | Ochem | 2188 | 1.32 | 0.32 |
| Ionization Potential | IP | Pubchem | 272 | 10.00 | 1.63 |
| Kovats Retention Index | KRI | Pubchem | 73507 | 2071.20 | 719.34 |
| Log P | LP | Pubchem | 28268 | 11.17 | 9.89 |
| Polarizability | POL | CCCB | 241 | 0.84 | 0.26 |
| Surface Tension | ST | Pubchem | 379 | 29.01 | 10.36 |
| Viscosity | VS | Pubchem | 294 | 0.47 | 0.87 |
| Heat of Vaporization | HV | Pubchem | 525 | 43.77 | 18.08 |

or extreme measurement conditions. All datasets were normalized using their respective means and standard deviations to ensure consistency during training.

From these datasets, we selected 23 source–target task pairs, considering the number of data points available in each dataset to maintain balance. Additionally, to ensure a fair and unbiased evaluation, we deliberately selected task pairs exhibiting a wide range of correlations, as illustrated in Figure 7.

Finally, we provide an explicit description of the physical meaning associated with each dataset to facilitate better understanding and context.

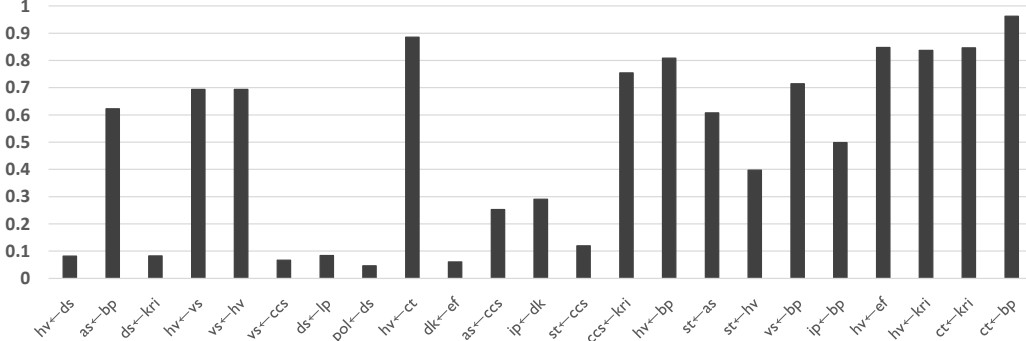

Figure 7: Pearson correlation between overlapping data points in target dataset and source dataset.

- **AS** : The solute dipolarity/polarizability.
- **BP** : The temperature at which this compound changes state from liquid to gas at a given atmospheric pressure.
- **CCS** : The effective area for the interaction between an individual ion and the neutral gas through which it is traveling.
- **CT** : The temparature when no gas can become liquid no matter how high the pressure is.
- **DK** : The ratio of the electric permeability of the material to the electric permeability of free space.

- **DS** : The mass of a unit volume of a compound.

- **EF** : The change in enthalpy resulting from the addition or removal of heat from 1 mole of a substance to change its state from a solid to a liquid.

- **IP** : The amount of energy required to remove an electron from an isolated atom or molecule.

- **KRI** : The rate at which a compound is processed through a gas chromatography column.

- **LP** : Logarithmic form of the ratio of concentrations of a compound in a mixture of octanol and water at equilibrium.

- **POL** : The tendency of matter, when subjected to an electric field, to acquire an electric dipole moment in proportion to that applied field.

- **ST** : The property of the surface of a liquid that allows it to resist an external force

- **VS** : A measure of a fluid's resistance to flow.

- **HV** : The quantity of heat that must be absorbed if a certain quantity of liquid is vaporized at a constant temperature.

### F.2 EXPERIMENTAL SETUPS

For the evaluation of GEAR, we compared its performance against seven benchmark models: GATE, STL, MTL, KD, global structure-preserving loss-based KD (GSP-KD), and transfer learning (either retraining the entire model or only the head network). All baselines share the same base architecture, with minor adjustments specific to each method.

GATE shares nearly identical network parameters with GEAR for the encoder and head networks. However, for the transfer module, GEAR requires maintaining the input and output vector dimensions across each layer. Accordingly, the hidden dimensions were adjusted to [50, 50, 50], [50,50,50] instead of [100, 100, 100], [100,100,100]. Other hyperparameters strictly follow those introduced in the original paper (Ko et al., 2023b).

In the MTL setup, the backbone and bottleneck layers are shared between the two tasks, while separate head networks are maintained for each task. For the KD baseline, latent vectors from the bottleneck are used as targets for knowledge distillation, with the distillation loss weighted at 0.1.

Graph Contrastive Representation Distillation (G-CRD) originally incorporates both contrastive and GSP losses (Joshi et al., 2022). However, since contrastive loss is unsuitable for regression tasks, we adopt only the GSP loss component. In GSP-KD, node features from the final layer of the backbone are used to compute pairwise distances, serving as the distillation targets. The loss ratio for GSP-based distillation is similarly set to 0.1.

Training is conducted for a maximum of 600 epochs, with the best model selected based on early stopping criteria.

## G EXPERIMENTAL RESULTS

We express explicit test results in this section. A total of 23 task pairs from 14 distinct datasets were thoroughly evaluated across eight different models. The full experimental results are presented across four tables. In each table, the best result for each task is highlighted with bold and underline, while the second-best result is underlined.

GEAR consistently outperforms other conventional methods by a significant margin. In both the random split and scaffold split settings, GEAR achieves the best performance on $73.91\%$ of the tasks. Furthermore, when considering both first and second place rankings, GEAR ranks within the top two for $95.65\%$ of all tasks.

Table 5: Random Split Result (part 1)

| Tasks | GEAR | | GATE | | STL | | MTL | |
|---|---|---|---|---|---|---|---|---|
| | RMSE | STD | RMSE | STD | RMSE | STD | RMSE | STD |
| **hv ← ds** | **0.8761** | 0.1145 | 0.9221 | 0.0612 | 0.9574 | 0.0519 | 0.9782 | 0.0782 |
| **as ← bp** | 0.4375 | 0.0188 | 0.4583 | 0.0193 | 0.5125 | 0.0085 | 0.4370 | 0.0119 |
| **ds ← kri** | **0.2796** | 0.0492 | 0.4145 | 0.0172 | 0.4154 | 0.0045 | 0.4172 | 0.0102 |
| **hv ← vs** | **0.5711** | 0.0358 | 0.9116 | 0.0522 | 0.9574 | 0.0519 | 0.9700 | 0.1052 |
| **vs ← hv** | **0.3364** | 0.0513 | 0.5471 | 0.0719 | 0.5947 | 0.0357 | 0.5535 | 0.0353 |
| **st ← as** | **0.6045** | 0.0981 | 0.6689 | 0.0413 | 0.9902 | 0.0729 | 1.0272 | 0.0244 |
| **ds ← lp** | **0.2677** | 0.0567 | 0.4046 | 0.0142 | 0.4154 | 0.0045 | 0.4133 | 0.0135 |
| **pol ← ds** | 0.2820 | 0.0362 | 0.3431 | 0.0475 | 0.3460 | 0.0291 | 0.4367 | 0.1213 |
| **vs ← bp** | **0.4299** | 0.0771 | 0.4457 | 0.0151 | 0.5947 | 0.0357 | 0.4516 | 0.0366 |
| **dk ← ef** | **0.3748** | 0.0092 | 0.4331 | 0.0140 | 0.4331 | 0.0358 | 0.4498 | 0.0126 |
| **as ← ccs** | **0.4400** | 0.0136 | 0.4648 | 0.0139 | 0.5125 | 0.0085 | 0.4677 | 0.0220 |
| **ct ← bp** | **0.1481** | 0.0138 | 0.1742 | 0.0034 | 0.2549 | 0.1247 | 0.1707 | 0.0132 |
| **st ← ccs** | **0.9222** | 0.0232 | 0.9546 | 0.0452 | 0.9902 | 0.0729 | 1.0361 | 0.0737 |
| **ccs ← kri** | 0.2426 | 0.0108 | 0.2476 | 0.0034 | 0.2936 | 0.0110 | 0.2524 | 0.0042 |
| **hv ← bp** | **0.6252** | 0.0320 | 0.7251 | 0.0581 | 0.9574 | 0.0519 | 0.7550 | 0.0432 |
| **vs ← ccs** | **0.3364** | 0.0513 | 0.5233 | 0.0323 | 0.5947 | 0.0357 | 0.5792 | 0.0228 |
| **st ← hv** | **0.5443** | 0.0841 | 0.7647 | 0.0622 | 0.9902 | 0.0729 | 0.7179 | 0.0259 |
| **hv ← ct** | **0.7481** | 0.1196 | 0.9399 | 0.0896 | 0.9574 | 0.0519 | 1.1118 | 0.1633 |
| **ip ← bp** | **0.4363** | 0.0307 | 0.5476 | 0.0642 | 0.6695 | 0.0660 | 0.6067 | 0.0345 |
| **hv ← ef** | 0.7409 | 0.1171 | **0.6131** | 0.0966 | 0.9574 | 0.0519 | 0.8296 | 0.0999 |
| **hv ← kri** | 0.6990 | 0.0888 | **0.5410** | 0.0732 | 0.9574 | 0.0519 | 0.8631 | 0.0354 |
| **ct ← kri** | **0.1481** | 0.0138 | 0.1658 | 0.0136 | 0.2549 | 0.1247 | 0.1716 | 0.0090 |
| **ip ← dk** | **0.5159** | 0.0362 | 0.6510 | 0.0381 | 0.6695 | 0.0660 | 0.7083 | 0.0226 |
| mean | **0.4785** | 0.0514 | 0.5592 | 0.0412 | 0.6642 | 0.0487 | 0.6263 | 0.0443 |
| | Count | Ratio | Count | Ratio | Count | Ratio | Count | Ratio |
| 1st | 18 | 78.26 % | 2 | 8.70 % | 0 | 0 % | 0 | 0 % |
| 2nd | 22 | 95.65 % | 13 | 56.52 % | 0 | 0 % | 2 | 8.70 % |

Table 6: Random Split Result (part 2)

| Tasks | KD RMSE | KD STD | GSP-KD RMSE | GSP-KD STD | Transfer All RMSE | Transfer All STD | Transfer Head RMSE | Transfer Head STD |
|---|---|---|---|---|---|---|---|---|
| hv ← ds | 1.3726 | 0.2930 | 0.9321 | 0.0487 | 1.0428 | 0.1165 | 1.1166 | 0.0024 |
| as ← bp | 0.5426 | 0.0335 | 0.5315 | 0.0151 | **0.4325** | 0.0104 | 0.7712 | 0.0105 |
| ds ← kri | 0.4403 | 0.0119 | 0.4147 | 0.0063 | 0.4414 | 0.0154 | 0.8842 | 0.0049 |
| hv ← vs | 1.1995 | 0.1419 | 0.9154 | 0.0130 | 0.9937 | 0.0821 | 1.0091 | 0.0181 |
| vs ← hv | 0.5878 | 0.0264 | 0.5619 | 0.0223 | 0.5712 | 0.0232 | 0.7215 | 0.0392 |
| st ← as | 1.1601 | 0.0396 | 0.9938 | 0.0141 | 1.1296 | 0.1302 | 1.0045 | 0.0220 |
| ds ← lp | 0.4378 | 0.0086 | 0.4106 | 0.0077 | 0.4280 | 0.0136 | 0.9111 | 0.0022 |
| pol ← ds | 0.3089 | 0.0270 | **0.2603** | 0.0270 | 0.3741 | 0.0303 | 0.9060 | 0.0141 |
| vs ← bp | 0.6076 | 0.0241 | 0.5932 | 0.0097 | 0.5445 | 0.0239 | 0.7220 | 0.0645 |
| dk ← ef | 0.3852 | 0.0238 | 0.4230 | 0.0133 | 0.3936 | 0.0164 | 0.9380 | 0.0026 |
| as ← ccs | 0.5364 | 0.0211 | 0.5457 | 0.0150 | 0.4741 | 0.0148 | 0.9935 | 0.0033 |
| ct ← bp | 0.1690 | 0.0079 | 0.2018 | 0.0093 | 0.1563 | 0.0044 | 0.6847 | 0.0186 |
| st ← ccs | 1.1731 | 0.0730 | 0.9595 | 0.0405 | 1.1334 | 0.0687 | 1.1039 | 0.0046 |
| ccs ← kri | 0.2622 | 0.0117 | 0.2698 | 0.0095 | **0.2273** | 0.0016 | 0.6166 | 0.0567 |
| hv ← bp | 1.1983 | 0.1815 | 0.9051 | 0.0571 | 0.8267 | 0.0417 | 0.8829 | 0.0499 |
| vs ← ccs | 0.6027 | 0.0127 | 0.5269 | 0.0167 | 0.4868 | 0.0119 | 0.8684 | 0.0116 |
| st ← hv | 1.1270 | 0.0184 | 0.9618 | 0.0086 | 1.0290 | 0.0945 | 1.0102 | 0.0138 |
| hv ← ct | 1.5114 | 0.1845 | 0.9207 | 0.0112 | 1.2072 | 0.0460 | 1.0302 | 0.0186 |
| ip ← bp | 0.5624 | 0.0273 | 0.4631 | 0.0037 | 0.9816 | 0.2334 | 0.8732 | 0.0293 |
| hv ← ef | 1.3659 | 0.2587 | 0.8112 | 0.0463 | 1.0818 | 0.1021 | 0.9616 | 0.0478 |
| hv ← kri | 1.3739 | 0.2487 | 0.9191 | 0.0676 | 0.9080 | 0.0510 | 1.0715 | 0.0145 |
| ct ← kri | 0.1586 | 0.0102 | 0.2080 | 0.0057 | 0.1661 | 0.0075 | 0.8349 | 0.0279 |
| ip ← dk | 0.5508 | 0.0100 | 0.5257 | 0.0192 | 0.6099 | 0.0273 | 1.0336 | 0.0085 |
| mean | 0.7667 | 0.0737 | 0.6198 | 0.0212 | 0.6800 | 0.0507 | 0.9108 | 0.0211 |
| mean | 0.7667 | 0.0737 | 0.6198 | 0.0212 | 0.6800 | 0.0507 | 0.9108 | 0.0211 |
|  | Count | Ratio | Count | Ratio | Count | Ratio | Count | Ratio |
| 1st | 0 | 0 % | 1 | 4.35 % | 2 | 8.70 % | 0 | 0 % |
| 2nd | 1 | 4.35 % | 4 | 17.39 % | 4 | 17.39 % | 0 | 0 % |

Table 7: Scaffold Split Result (part 1)

| Tasks | GEAR RMSE | STD | GATE RMSE | STD | STL RMSE | STD | MTL RMSE | STD |
|---|---|---|---|---|---|---|---|---|
| **hv ← ds** | 0.6101 | 0.0210 | 0.6939 | 0.0996 | 0.6744 | 0.1079 | 0.6465 | 0.0776 |
| **as ← bp** | **1.0016** | 0.0073 | 1.0495 | 0.0256 | 1.2828 | 0.0724 | 1.1677 | 0.1068 |
| **ds ← kri** | **0.4261** | 0.0017 | 0.4395 | 0.0108 | 0.4477 | 0.0052 | 0.4849 | 0.0061 |
| **hv ← vs** | **0.5731** | 0.0470 | 0.7174 | 0.0796 | 0.6744 | 0.1079 | 0.9954 | 0.2059 |
| **vs ← hv** | 0.6323 | 0.0441 | **0.6120** | 0.0639 | 0.9816 | 0.1267 | 0.8535 | 0.0558 |
| **st ← as** | **0.6980** | 0.0832 | 0.7540 | 0.0660 | 0.8041 | 0.1062 | 1.0254 | 0.0251 |
| **ds ← lp** | 0.4236 | 0.0036 | **0.4049** | 0.0102 | 0.4477 | 0.0052 | 0.4517 | 0.0184 |
| **pol ← ds** | 0.9902 | 0.0697 | 0.9040 | 0.0852 | 0.9604 | 0.1056 | 1.4198 | 0.0796 |
| **vs ← bp** | **0.5242** | 0.0418 | 0.6121 | 0.0297 | 0.9816 | 0.1267 | 0.5686 | 0.0276 |
| **dk ← ef** | **0.5229** | 0.0166 | 0.7122 | 0.0545 | 0.7028 | 0.0391 | 0.6549 | 0.0210 |
| **as ← ccs** | **1.0016** | 0.0073 | 1.1313 | 0.0496 | 1.2828 | 0.0724 | 1.1197 | 0.0558 |
| **ct ← bp** | **0.3275** | 0.0329 | 0.3883 | 0.0203 | 1.4436 | 0.1150 | 0.4359 | 0.0126 |
| **st ← ccs** | **0.6975** | 0.0833 | 0.7281 | 0.0586 | 0.8041 | 0.1062 | 0.9905 | 0.0737 |
| **ccs ← kri** | **0.5111** | 0.0044 | 0.5292 | 0.0094 | 0.5489 | 0.0107 | 0.5297 | 0.0083 |
| **hv ← bp** | 0.4671 | 0.0136 | 0.4821 | 0.0132 | 0.6744 | 0.1079 | **0.4668** | 0.0169 |
| **vs ← ccs** | **0.5611** | 0.0676 | 0.6126 | 0.0671 | 0.9816 | 0.1267 | 0.8186 | 0.0790 |
| **st ← hv** | **0.6980** | 0.0832 | 0.7209 | 0.0412 | 0.8041 | 0.1062 | 0.7237 | 0.0276 |
| **hv ← ct** | **0.5038** | 0.0236 | 0.6579 | 0.0678 | 0.6744 | 0.1079 | 0.6633 | 0.0660 |
| **ip ← bp** | **0.4064** | 0.0300 | 0.4668 | 0.0179 | 0.5780 | 0.1475 | 0.5540 | 0.0587 |
| **hv ← ef** | **0.5038** | 0.0236 | 0.6406 | 0.0335 | 0.6744 | 0.1079 | 0.7879 | 0.0643 |
| **hv ← kri** | **0.4812** | 0.0150 | 0.5084 | 0.0264 | 0.6744 | 0.1079 | 0.6204 | 0.0269 |
| **ct ← kri** | 0.4256 | 0.0214 | **0.3902** | 0.0140 | 1.4436 | 0.1150 | 0.5173 | 0.0927 |
| **ip ← dk** | **0.3984** | 0.0232 | 0.4335 | 0.0119 | 0.5780 | 0.1475 | 0.5335 | 0.1016 |
| mean | **0.5820** | 0.0333 | 0.6343 | 0.0416 | 0.8313 | 0.0949 | 0.7404 | 0.0569 |

| | Count | Ratio | Count | Ratio | Count | Ratio | Count | Ratio |
|---|---|---|---|---|---|---|---|---|
| 1st | 17 | 73.91 % | 3 | 13.04 % | 0 | 0 % | 1 | 4.35 % |
| 2nd | 22 | 95.65 % | 14 | 60.87 % | 0 | 0 % | 3 | 13.04 % |

Table 8: Scaffold Split Result (part 2)

| Tasks | KD RMSE | KD STD | GSP-KD RMSE | GSP-KD STD | Transfer All RMSE | Transfer All STD | Transfer Head RMSE | Transfer Head STD |
|---|---|---|---|---|---|---|---|---|
| hv ← ds | **0.5920** | 0.0466 | 0.7606 | 0.0810 | 0.8659 | 0.0788 | 0.9584 | 0.0339 |
| as ← bp | 1.3580 | 0.0136 | 1.2340 | 0.0294 | 1.1478 | 0.0264 | 1.0935 | 0.0079 |
| ds ← kri | 0.5409 | 0.0480 | 0.4467 | 0.0104 | 0.8753 | 0.1134 | 1.0928 | 0.0482 |
| hv ← vs | 0.8948 | 0.2294 | 0.6536 | 0.0345 | 0.7520 | 0.1666 | 0.7924 | 0.0595 |
| vs ← hv | 1.2597 | 0.3638 | 0.6377 | 0.0253 | 0.9217 | 0.1575 | 0.9179 | 0.0539 |
| st ← as | 1.7083 | 0.1608 | 0.9335 | 0.0954 | 1.2604 | 0.0946 | 1.0780 | 0.0613 |
| ds ← lp | 0.5221 | 0.0328 | 0.4685 | 0.0111 | 0.4664 | 0.0121 | 1.0410 | 0.0026 |
| pol ← ds | 1.3309 | 0.1998 | **0.8475** | 0.0627 | 1.0385 | 0.2146 | 1.3204 | 0.0491 |
| vs ← bp | 0.9371 | 0.2386 | 0.6599 | 0.0204 | 1.1532 | 0.1766 | 1.0135 | 0.0820 |
| dk ← ef | 0.8189 | 0.0462 | 0.6353 | 0.0171 | 0.7417 | 0.0384 | 0.7963 | 0.0071 |
| as ← ccs | 1.3773 | 0.0781 | 1.1272 | 0.0778 | 1.2925 | 0.0606 | 1.4530 | 0.0143 |
| ct ← bp | 1.2459 | 0.1199 | 1.1837 | 0.0586 | 0.5644 | 0.0530 | 0.9347 | 0.0316 |
| st ← ccs | 1.5402 | 0.1418 | 0.7344 | 0.0187 | 0.9075 | 0.0431 | 1.2596 | 0.0287 |
| ccs ← kri | 0.5534 | 0.0190 | 0.5356 | 0.0115 | 0.5640 | 0.0137 | 0.7904 | 0.0159 |
| hv ← bp | 0.6271 | 0.0868 | 0.7403 | 0.0889 | 0.6093 | 0.0422 | 0.8111 | 0.0251 |
| vs ← ccs | 1.3034 | 0.5354 | 0.8027 | 0.0159 | 0.7271 | 0.0828 | 1.2282 | 0.0243 |
| st ← hv | 1.5256 | 0.1906 | 0.7417 | 0.0206 | 1.4243 | 0.0627 | 1.0047 | 0.0813 |
| hv ← ct | 0.7925 | 0.2694 | 0.6428 | 0.0080 | 0.9499 | 0.2579 | 0.8089 | 0.0532 |
| ip ← bp | 0.4205 | 0.0240 | 0.4579 | 0.0207 | 0.4419 | 0.0371 | 0.9704 | 0.0399 |
| hv ← ef | 0.6773 | 0.1553 | 0.5862 | 0.0375 | 1.0003 | 0.1719 | 0.9503 | 0.0307 |
| hv ← kri | 0.6710 | 0.1524 | 0.5509 | 0.0252 | 0.6560 | 0.0408 | 0.9998 | 0.0311 |
| ct ← kri | 1.3392 | 0.1076 | 1.2358 | 0.0373 | 1.1124 | 0.1265 | 1.2769 | 0.0193 |
| ip ← dk | 0.4975 | 0.0769 | 0.4376 | 0.0255 | 0.5248 | 0.0471 | 1.0165 | 0.0521 |
| mean | 0.9797 | 0.1451 | 0.7415 | 0.0362 | 0.8694 | 0.0921 | 1.0265 | 0.0217 |
| | Count | Ratio | Count | Ratio | Count | Ratio | Count | Ratio |
| 1st | 1 | 4.35 % | 1 | 4.35 % | 0 | 0 % | 0 | 0 % |
| 2nd | 2 | 8.70 % | 5 | 21.74 % | 0 | 0 % | 0 | 0 % |

# H ADDITIONAL ABLATION STUDY

## H.1 SENSITIVITY TO LATENT SPACE DIMENSIONALITY

In this section, to further assess the robustness of our method, we conducted an ablation study examining how the latent space dimensionality—determined by the transfer and inverse transfer modules—affects performance. As shown in Figure 8, we varied the latent dimension across 5, 20, 50, and 200 on the HV prediction task using the EF task as the source(hv ← ef), and tracked the corresponding validation loss trends. We observed that the validation loss consistently decreased as the latent dimension increased up to 50, and then rose slightly beyond that point. Notably, the differences in validation loss among dimensions 20, 50, and 200 were minimal, indicating that—as long as the latent space is not excessively small—GEAR exhibits stable training behavior.

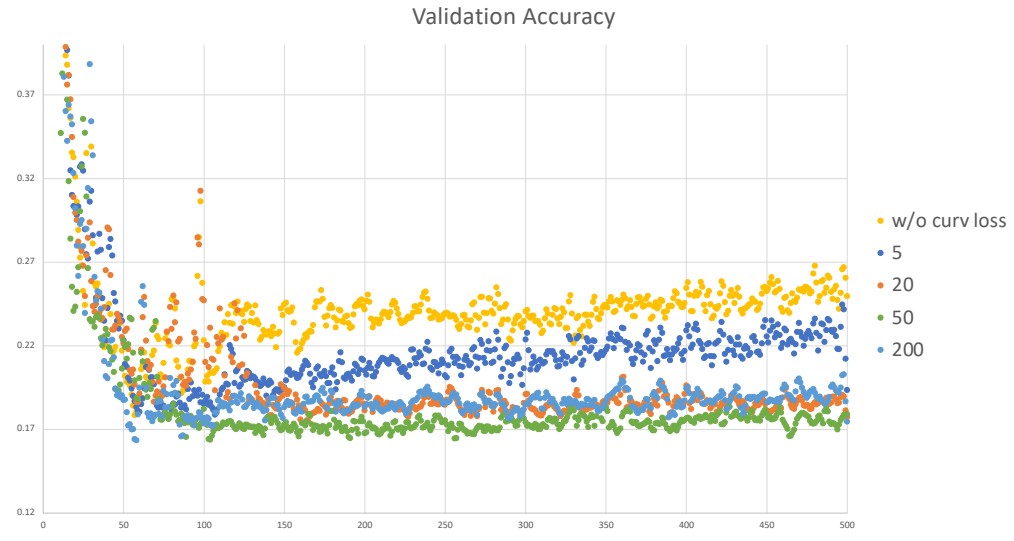

Figure 8: This figure depicts validation accuracy of different latent space dimensions.

## H.2 OPTIMIZATION OF LOSS WEIGHTS

We adopted a two-stage procedure for hyperparameter optimization. Firstly, we initialized all loss-weight hyperparameters to comparable magnitudes ($\alpha = \beta = \gamma = 1$, $\delta = \epsilon = 0.1$). This avoids early imbalance among loss components.

Then, we performed grid search on validation set. A grid over 0.1, 0.2, 0.5, 1 was used for $\alpha$, $\beta$, $\gamma$, $\delta$, and $\epsilon$, total 1024 combinations. The optimal values were chosen based on average validation RMSE across 23 task pairs. This same procedure was applied consistently to all tasks. The results are shown in table 9.

| Task | Minimum RMSE | Mean RMSE | Std. Dev. | Ours |
|---|---|---|---|---|
| AS → Target | 0.3957 | 0.4314 | 0.0136 | 0.4033 |
| ST → Target | 0.1746 | 0.1972 | 0.0141 | 0.1764 |

Table 9: Hyperparameter sensitivity analysis for AS→Target and ST→Target validation tasks.

Although the precise optimum varies per task, the selected setting is highly stable: it is near-optimal for AS and ST, and was empirically found to be the most robust across all 23 tasks. The variation across the grid is modest, indicating that GEAR does not rely on fragile or highly tuned hyperparameters.

Finally, we emphasize that the complete ablation sweep for both target tasks required only approximately 1400 seconds (under 30 minutes) on 8 x A40 GPUs, demonstrating that the hyperparameter analysis is computationally lightweight and practical. These results demonstrate that GEAR is stable under reasonable hyperparameter variations.

### H.3 REPLACEMENT OF CURVATURE LOSS TO HESSIAN-BASED LOSS

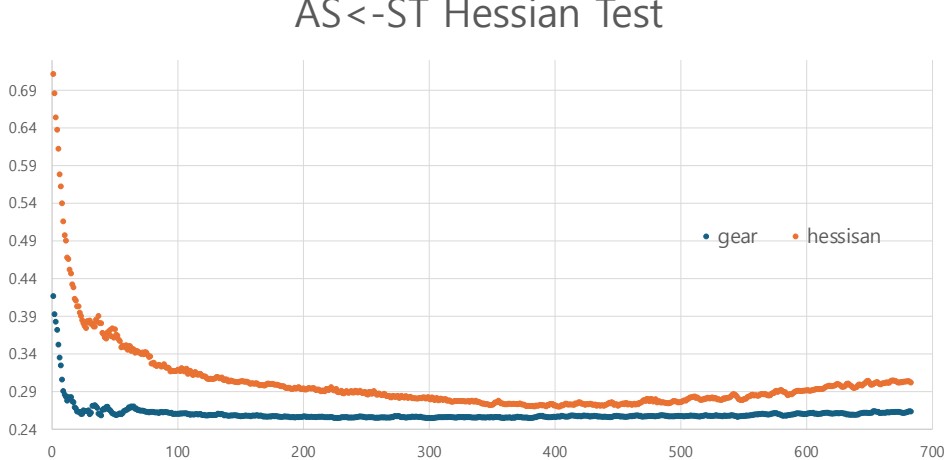

Figure 9: This plot shows the loss curve when curvature loss is replaced with a Hessian-based loss. The validation RMSE curves clearly show that, unlike curvature, the Hessian loss fails to guide learning: performance is inferior and exhibits severe overfitting.

## I THE USE OF LARGE LANGUAGE MODELS (LLMs)

We employed large language models to refine the grammar and improve the clarity of the text.

