# OpenReview forum: "Geometric Embedding Alignment via Curvature Matching in Transfer Learning"
_ICLR.cc/2026/Conference — ICLR 2026 Conference Withdrawn Submission_

### Official Review · Reviewer_6FvH · 2025-10-27

**Soundness:** 2
**Presentation:** 3
**Contribution:** 2
**Rating:** 2
**Confidence:** 4

**Summary:**

This work proposes GEAR, a transfer learning framework that uses Riemannian geometry to align the Ricci curvature of latent spaces across multiple models. Evaluations are performed to show effectiveness.

**Strengths:**

a) The paper is well organized.

**Weaknesses:**

**(a) Limited novelty**
The proposed method appears to build heavily on GATE, with Fig. 1 in this paper closely resembling Fig. 2 from GATE. The new approach, GEAR, seems to primarily add an additional term to GATE rather than introducing fundamentally new ideas. This raises concerns about the extent of technical novelty and distinct contribution.

**(b) Poor readability**
Sections 3.1 and 3.2 contain extensive derivative formulations that appear unnecessary for understanding the method and make the paper harder to follow. Many of these formulas could be moved to the appendix to reduce cognitive load for readers. In practice, the most relevant parts are the loss definitions — which again are similar to those used in GATE.

**(c) Limited applicability**
The proposed transfer learning strategy is presented as a generic framework, yet its evaluation is restricted to molecular property prediction tasks. Without experiments on a broader range of domains, the generality and potential impact of the method outside the molecular setting remain unclear.

**(d) Insufficient model details**
The paper does not clearly describe the backbone network architectures used for the tasks. For a claimed general-purpose algorithm, it would be important to validate performance across more model architectures and report these details explicitly.

**(e) Incomplete ablation analysis**
The overall loss function contains six distinct terms with associated weighting parameters, making training potentially complex and sensitive to hyperparameter choice. The paper lacks analysis of how these weighting parameters influence performance or how each loss component contributes to the final results. A thorough ablation would be necessary to understand the method’s robustness.

**Questions:**

See weakness.

---

> ### Author Response · Authors · 2025-11-17
>
> Before reading the following comments, we kindly suggest that the reviewer refer to the summary above for a concise overview of our method.
>
> The reviewer may have overlooked key distinctions in our article because GATE and GEAR share the high-level motivation of geometric alignment for transfer learning. However, although the conceptual motivation is related, the construction of the model is fundamentally different. Below we address each weakness in detail.
>
> **(a) Limited Novelty**
>
> We appreciate the concern regarding overlap with GATE and acknowledge that GEAR intentionally adopts a modular design similar to GATE so that the effect of adding geometric reasoning can be cleanly isolated. However, GEAR introduces several substantial and nontrivial innovations that go well beyond “adding an additional term”:
>
> 1. Introduction of the analytic Riemannian metric and local-flat-space construction
> GEAR derives an explicit Riemannian metric from the Jacobian of the transfer module and enforces local flatness in the shared space via the metric loss.
> This component does not exist in GATE.
> GATE performs perturbation-based distance matching, which only approximates first-order geometry and has no formal metric structure.
>
> 2. Analytic computation of the Ricci scalar curvature
> GEAR defines curvature analytically from the Jacobian and its derivatives, allowing for:
> - closed-form curvature expressions,
>
> - explicit curvature supervision,
>
> - a truly global geometric signal.
>
> GATE has no concept of curvature; it operates purely via sampled finite differences in the original latent space.
>
> 3. Curvature matching as a transfer principle
> Curvature is an intrinsic, diffeomorphism-invariant descriptor of manifold geometry.
> By aligning Ricci curvature in a shared flat coordinate frame, GEAR matches global geometric structure, whereas GATE aligns only local perturbations.
>
>
> This is a fundamentally different—and mathematically stronger—alignment paradigm.
>
> 4. New geometric losses: the metric loss and curvature loss
> GATE uses three losses (reconstruction, mapping, contrastive distance).
> GEAR replacing distance loss and perturbations with two new geometric constraints with clear theoretical motivation:
>
> - metric loss (enforces local flatness and stabilizes differential structure)
>
> - curvature loss (aligns global manifold structure)
>
> 5. Architectural Differences
> A key limitation of GATE is that its perturbation-based distance matching requires perturbation points to lie on the same manifold. Consequently, all tasks must share an identical embedding architecture, restricting flexibility and preventing the use of modality-specific or task-specific encoders.
> In contrast, GEAR extends geometric alignment from local perturbation matching to curvature-based matching, which does not require perturbation points. Because curvature is computed analytically via the Jacobian-induced metric, GEAR no longer depends on shared perturbations or identical encoders. Each task may freely use its own embedding architecture.
>
> This represents a substantial architectural improvement over GATE.
> GEAR not only achieves superior performance across many experiments, but also removes the architectural coupling constraint, enabling extensions to heterogeneous modalities and domains (e.g., vision, NLP, multimodal learning).
>
> 6. Empirical consequences
> GEAR yields:
>
> - significantly improved robustness under label noise,
>
> - improved performance in low-data regimes,
>
> - smoother optimization dynamics (Fig. 3),
>
> - more stable transfer even for low-correlation task pairs.
>
> Thus, while GEAR inherits the modular structure of GATE, the underlying mathematics, loss structure, and geometric alignment mechanism represent substantial new contributions.

---

> ### Author Response · Authors · 2025-11-17
>
> **(b) Readability and mathematical density**
>
> We agree that Sections 3.1–3.2 contain substantial mathematical detail. These derivations were included to ensure full transparency for readers familiar with differential geometry. For example, Section 3.1 introduces the definition of the Ricci scalar curvature that our method targets, while Section 3.2 presents the key components required to compute this quantity analytically. The formulas are intentionally minimal, and every term shown is essential to the method.
>
> We appreciate the reviewer’s concern that the differential-geometric framework may be challenging for some readers. In the manuscript, we aimed to provide a mathematically precise treatment of metrics and curvature while also offering a pedagogical introduction for readers with less background in geometry. However, it is inherently difficult to convey both the intuition and the formalism of differential geometry without presenting explicit equations.
>
> Because our method is fundamentally grounded in Riemannian geometry, a certain level of mathematical detail is unavoidable. We have attempted to balance rigor and readability, but further simplifying the theoretical material would compromise the correctness and interpretability of our approach.
>
> To support readers, Appendices A and B introduce the required notation and definitions, and Appendix C provides full derivations. These materials are designed to make the geometric ideas accessible while maintaining the necessary level of technical rigor.
>
> **(c) Limited applicability**
>
> We agree that demonstrating broader applicability is important. In this submission, we focused on molecular property regression tasks because:
>
> - Our primary ICLR area is **applications to physical sciences (physics, chemistry, biology, etc.)**,
>
> - And we wanted a controlled comparison against the GATE framework on exactly the same benchmark setting.
>
> That said, the proposed GEAR framework is modality-agnostic and task-agnostic by design:
>
> - The only domain-specific component is the encoder that produces the initial embeddings (DMPNN for molecules in our experiments).
>
> - All subsequent components—encoder bottleneck, transfer module, inverse transfer module, metric and curvature computation, and the curvature/mapping/consistency losses—only operate on latent vectors, not on molecular features specifically.
>
> As we note in our responses and discussion section:
>
> - For classification tasks, one simply replaces the regression head with a classification head; the curvature and mapping losses, metric loss, and alignment mechanism remain unchanged.
>
> - For other modalities (vision, NLP, graphs, multimodal data), any encoder that outputs continuous latent vectors (e.g., ResNet-50, InceptionV3, BERT, GNNs) can be plugged in, and GEAR can then align the latent geometries across tasks exactly as in the molecular case.
>
> We are planning to extend GEAR to additional modalities and domains in future work
>
>
> **(d) Insufficient model details**
>
> All architecture specifications—including:
>
> - DMPNN embedding module,
>
> - encoder bottleneck dimensions,
>
> - decoder dimensions,
>
> - transfer/inverse-transfer module MLP sizes,
>
> - activation functions, etc...
>
> are given in Appendix E, Tables 1–3 as it is addressed in the main contents.
>
> However, for the backbone architecture (DMPNN), which is already described in Appendix D, we will explicitly reference it in the main text in the revised version to improve clarity.

---

> ### Author Response · Authors · 2025-11-17
>
> **(e) Incomplete ablation analysis**
>
> We appreciate the reviewer emphasizing the importance of analyzing the effect of the five hyperparameters (alpha, beta, gamma, delta, epsilon). We will explicitly add tuning protocol in the appendix H.
>
> **Practical tuning strategy**
> We adopted a two-stage procedure:
>
> 1. Initialization with equal scale.
> All loss-weight hyperparameters were initialized to comparable magnitudes
> (𝛼 = 𝛽 = 𝛾 = 1, δ = ϵ = 0.1).
> This avoids early imbalance among loss components.
>
> 2. Grid search on validation sets.
> A grid over
> {0.1, 0.2, 0.5, 1} was used for α, β, γ, δ, ϵ.
> The optimal values were chosen based on average validation RMSE across 23 task pairs.
> This same procedure was applied consistently to all tasks.
>
> **Sensitivity analysis**
> We performed a full sensitivity sweep using the same grid.
>
> For the AS→target validation task:
>
> - Minimum RMSE: 0.3957
>
> - Mean RMSE across all combinations: 0.4314
>
> - Standard deviation: 0.0136
>
> - Ours (chosen configuration): 0.4033
>
> For ST → target validation task:
>
> - Minimum RMSE: 0.1746
>
> - Mean RMSE across all combinations: 0.1972
>
> - Standard deviation: 0.0141
>
> - Ours (chosen configuration): 0.1764
>
>
> Although the precise optimum varies per task, the selected setting is highly stable:
> it is near-optimal for AS and ST, and was empirically found to be the most robust across all 23 tasks.
> The variation across the grid is modest, indicating that GEAR does not rely on fragile or highly tuned hyperparameters.
>
> Finally, we emphasize that the complete ablation sweep for both target tasks required only ≈1400 seconds (under 30 minutes) on 8× A40 GPUs, demonstrating that the hyperparameter analysis is computationally lightweight and practical.
>
> **Conclusion**
> These results demonstrate that GEAR is stable under reasonable hyperparameter variations.
> We will include the AS, ST-task sensitivity table and a brief discussion of this analysis in the revised manuscript.
>
> We believe these responses address the reviewer’s concerns and will substantially improve the clarity and perceived contribution of the work.

---

### Official Review · Reviewer_oNjo · 2025-10-30

**Soundness:** 3
**Presentation:** 3
**Contribution:** 3
**Rating:** 6
**Confidence:** 2

**Summary:**

This paper introduces GEAR, a novel transfer-learning framework designed to enhance the robustness and efficiency of knowledge transfer, particularly for data-scarce regression tasks.

The work is motivated by a critique of existing geometric transfer-learning methods. The authors posit that current approaches are limited by a "local" and myopic alignment strategy, which focuses on matching infinitesimal distances within the latent space. This, they argue, fails to capture and align the intrinsic global geometric structure of the latent spaces, thereby hindering the potential for fundamentally robust and efficient knowledge transfer.

To address this perceived limitation, the paper proposes GEAR, a framework grounded in Riemannian differential geometry. The architecture is built upon three core components: (1) independent task encoders that map source and target data into their respective curved latent-space manifolds; (2) a central geometric transfer module that acts as a bridge, mapping these distinct manifolds onto a shared, globally flat common manifold; and (3) a curvature-matching function that serves as the guiding principle for the framework. This loss term is designed to analytically compute and align the Ricci scalar curvature of the two latent manifolds, steering the transfer process.

**Strengths:**

1. The method proposed in this paper apply the idea of matching the Ricci scalar curvature from Riemannian differential geometry to the problem of transfer learning. Compared with prior approaches that rely on local perturbation-based alignment, GEAR captures and aligns the latent space’s global geometric structure by matching curvature.

2. The authors conduct extensive comparisons of GEAR against multiple baseline models and demonstrate its superior performance. Detailed ablation studies clearly validate the crucial role of key model components—particularly the curvature loss—in mitigating overfitting and improving performance.

**Weaknesses:**

1. The discussion of the loss-function design is insufficient, particularly with respect to sensitivity analysis for the multiple hyperparameters. The final total loss (Equation 23) comprises five terms that must be balanced and is controlled by five hyperparameters (α, β, γ, δ, ε). Such a complex design requires careful tuning, yet the paper provides no analysis of how variations in these hyperparameters affect model performance. A method that is highly sensitive to hyperparameters raises questions about its robustness and generalization in practical applications. I therefore recommend that the authors include, in the revised manuscript, a description of the hyperparameter tuning procedure and a sensitivity analysis for these key hyperparameters to demonstrate the practicality and stability of their approach.
2. The paper’s theoretical motivation is built on the assumption of a “shared manifold M,” i.e., that the latent spaces of different tasks are mapped from a common underlying manifold. However, in the model’s concrete implementation this shared manifold M is not explicitly parameterized or constructed. The alignment operation is realized by computing curvature within each latent space and imposing a loss that forces these curvature measures to match. This alignment appears to occur in a shared locally flat framework rather than on the theoretical manifold M, producing a gap between theory and practice: does matching pointwise-computed Ricci curvature alone guarantee that two latent spaces are effectively aligned onto a unified, coherent underlying manifold? The authors should clearly explain the practical role and instantiation (if any) of the shared manifold M in their implementation.


Minor revision:
1. In Figure 2, the annotated RMSE (root mean square error) is a “lower is better” metric, but it is currently labeled as if “higher is better.”

**Questions:**

1. The loss function involves five hyperparameters, but the paper lacks a sensitivity analysis, raising questions about the method's practical robustness. Could you provide an analysis of how performance is affected by these hyperparameters to demonstrate the stability of your approach?

2. The theory is based on a "shared manifold M," but the implementation appears to only match Ricci curvatures. Could you clarify how matching curvature practically achieves alignment onto this theoretical shared manifold and why this is a sufficient condition?

---

> ### Author Response · Authors · 2025-11-17
>
> Before reading the following comments, we kindly suggest that the reviewer refer to the summary above for a concise overview of our method.
>
> We thank the reviewer for the positive assessment of our contribution, the clear recognition of the novelty of using Ricci scalar curvature for transfer learning, and the thoughtful comments regarding the theoretical and practical aspects of GEAR. We address the concerns in detail below.
>
> **1. Hyperparameter Sensitivity and Tuning Procedure**
>
> We appreciate the reviewer emphasizing the importance of analyzing the effect of the five hyperparameters (alpha, beta, gamma, delta, epsilon). We will explicitly add tuning protocol in the appendix H.
>
> **Practical tuning strategy**
> We adopted a two-stage procedure:
>
> 1. Initialization with equal scale.
> All loss-weight hyperparameters were initialized to comparable magnitudes
> (𝛼 = 𝛽 = 𝛾 = 1, δ = ϵ = 0.1).
> This avoids early imbalance among loss components.
>
> 2. Grid search on validation sets.
> A grid over
> {0.1, 0.2, 0.5, 1} was used for α, β, γ, δ, ϵ.
> The optimal values were chosen based on average validation RMSE across 23 task pairs.
> This same procedure was applied consistently to all tasks.
>
> **Sensitivity analysis**
> We performed a full sensitivity sweep using the same grid.
>
> For the AS→target validation task:
>
> - Minimum RMSE: 0.3957
>
> - Mean RMSE across all combinations: 0.4314
>
> - Standard deviation: 0.0136
>
> - Ours (chosen configuration): 0.4033
>
> For ST → target validation task:
>
> - Minimum RMSE: 0.1746
>
> - Mean RMSE across all combinations: 0.1972
>
> - Standard deviation: 0.0141
>
> - Ours (chosen configuration): 0.1764
>
>
> Although the precise optimum varies per task, the selected setting is highly stable:
> it is near-optimal for AS and ST, and was empirically found to be the most robust across all 23 tasks.
> The variation across the grid is modest, indicating that GEAR does not rely on fragile or highly tuned hyperparameters.
>
> Finally, we emphasize that the complete ablation sweep for both target tasks required only ≈1400 seconds (under 30 minutes) on 8× A40 GPUs, demonstrating that the hyperparameter analysis is computationally lightweight and practical.
>
> **Conclusion**
> These results demonstrate that GEAR is stable under reasonable hyperparameter variations.
> We will include the AS, ST-task sensitivity table and a brief discussion of this analysis in the revised manuscript.
>
> **2. Practical Interpretation of the “Shared Manifold M”**
>
> We thank the reviewer for highlighting this conceptual question. Below we clarify how the shared manifold is realized in practice.
>
> (a) Theoretical assumption
>
> The theory assumes that the latent spaces of the source and target tasks correspond to charts on a shared underlying manifold M.
> This is a standard assumption in geometry-based transfer learning (including GATE).
>
> (b) Practical instantiation in GEAR
>
> Importantly, GEAR does not explicitly construct or parameterize M.
> Instead, the shared manifold is realized implicitly through the geometry of the transfer module.
>
> Key idea:
> If two Riemannian manifolds $(M_s, g_s)$ and $(M_t, g_t)$ have matching metrics and matching Ricci scalar curvature in a common coordinate frame, then there exists a smooth, locally diffeomorphic map aligning them onto the same underlying manifold.
>
> GEAR builds such a shared frame as follows:
>
> (1) Both latent spaces are mapped into a common locally flat reference frame via the transfer module.
> This ensures that their metrics approximately match a flat metric, achieved through the metric loss.
>
> (2) Ricci scalar curvature is then analytically computed in this common frame from the Jacobian of the transfer mapping.
>
> (3) The curvature loss aligns the global intrinsic geometry by matching these curvature scalars across tasks.
>
> This process avoids the need to explicitly construct M:
>
> - The shared coordinate frame behaves as a chart of the underlying manifold.
>
> - Matching curvature ensures that both tasks share the same intrinsic geometric structure.
>
> - The transfer mapping acts as the implicit diffeomorphism “gluing” the two charts together.

---

> ### Author Response · Authors · 2025-11-17
>
> (c) Why curvature matching is sufficient
>
> We agree with the reviewer that matching scalar curvature alone is not sufficient to guarantee full manifold alignment. In GEAR, curvature matching is only one component of a broader geometric alignment mechanism. The key point is that GEAR aligns manifolds through the combined effect of:
>
> - Curvature loss
>
> - Metric loss
>
> - Consistency loss
>
> These losses work together to ensure that curvature is computed correctly and that the source and target manifolds are coherently aligned onto a shared latent manifold 𝑀
>
> (1) Metric loss: ensuring a shared locally-flat coordinate system
>
> Ricci curvature is well defined only with respect to a consistent metric.
> The metric loss forces both tasks—after mapping through the transfer module—to adopt a locally flat coordinate system on
> 𝑀.
>
> This is grounded in a standard fact from Riemannian geometry:
> any curved manifold admits a coordinate chart in which the metric becomes Euclidean at a point (normal coordinates).
>
> By encouraging the latent metric to match the Euclidean metric in this shared coordinate frame, the metric loss ensures that:
>
> - both tasks use compatible coordinates,
>
> - curvature can be computed consistently,
>
> - the induced geometries are made comparable on 𝑀.
>
> (2) Consistency loss: aligning corresponding points on 𝑀
>
> The consistency loss ensures that corresponding latent vectors from different tasks map to the same point in the shared manifold:
>
> $\Phi_s(z_s) \sim \Phi_t(z_t)$.
>
> This effectively “glues’’ the charts together so curvature and metric are evaluated on the same manifold location, rather than misaligned embeddings.
>
> Curvature matching: aligning global geometry
>
> Once consistency and local-flatness constraints are satisfied, the curvature loss can be applied meaningfully.
> Matching Ricci scalar curvature in this frame aligns the global intrinsic geometric structure of the two task manifolds.
>
> In GEAR, the latent-space geometry is induced by the Jacobian 𝐽 of the transfer module:
>
> $g = J^TJ$.
>
> Thus, curvature matching constrains the Jacobian, ensuring that the induced geometries from both tasks coincide on the shared manifold 𝑀.
>
>
>
>
> **3. Minor: RMSE orientation in Figure 2**
>
> As it is described in the figure 2, each axis plots the GEAR RMSE divided by the benchmark model RMSE.
>
> (GEAR RMSE) / (benchmark model RMSE)
>
> which makes “higher is better.”

---

### Official Review · Reviewer_uBNt · 2025-11-01

**Soundness:** 2
**Presentation:** 2
**Contribution:** 3
**Rating:** 4
**Confidence:** 3

**Summary:**

The paper proposes a transfer learning algorithm based on tools from differential geometry, specifically Ricci curvature, for molecular property prediction (regression task). The main idea is to align the latent spaces of a source and a target task by learning mappings into a shared, locally flat frame, and matching their Ricci scalar curvature. The proposed algorithm is based on multiple loss functions, including mapping, curvature, and autoencoder reconstruction losses. The authors evaluate their method on 23 molecular task pairs across 14 molecular properties.

**Strengths:**

- I think the use of Ricci curvature for global latent space alignment is a novel and interesting contribution to the field of transfer learning and molecular property prediction.

- The authors support their idea with an ablation study showing that the proposed curvature and mapping losses are important for model performance, as removing them could lead to overfitting.

**Weaknesses:**

* The experiments are mainly focused on molecular property prediction. It would be valuable to see how these results extend to other domains/ tasks and data modalities.
* The presentation of the paper could be improved further, as the underlying math is sometimes advanced, which might limit the accessibility of the method to a broader audience.
* The paper focuses on a two-task setting (source and target). The authors claim the framework is extensible to scenarios with more than two tasks, but this is not shown. It is unclear, e.g., how the curvature matching would be managed in a multi-task setting.

**Questions:**

See above.

---

> ### Author Response · Authors · 2025-11-17
>
> Before reading the following comments, we kindly suggest that the reviewer refer to the summary above for a concise overview of our method.
>
> We thank the reviewer for the thoughtful assessment, the positive evaluation of our contribution, and the constructive suggestions regarding scope, clarity, and the multi-task setting. We address each point below.
>
> **1. Scope beyond molecular property prediction / other modalities and tasks**
>
> We agree that demonstrating broader applicability is important. In this submission, we focused on molecular property regression tasks because:
>
> - Our primary ICLR area is **applications to physical sciences (physics, chemistry, biology, etc.)**,
>
> - And we wanted a controlled comparison against the GATE framework on exactly the same benchmark setting.
>
> That said, the proposed GEAR framework is modality-agnostic and task-agnostic by design:
>
> - The only domain-specific component is the encoder that produces the initial embeddings (DMPNN for molecules in our experiments).
>
> - All subsequent components—encoder bottleneck, transfer module, inverse transfer module, metric and curvature computation, and the curvature/mapping/consistency losses—only operate on latent vectors, not on molecular features specifically.
>
> As we note in our responses and discussion section:
>
> - For classification tasks, one simply replaces the regression head with a classification head; the curvature and mapping losses, metric loss, and alignment mechanism remain unchanged.
>
> - For other modalities (vision, NLP, graphs, multimodal data), any encoder that outputs continuous latent vectors (e.g., ResNet-50, InceptionV3, BERT, GNNs) can be plugged in, and GEAR can then align the latent geometries across tasks exactly as in the molecular case.
>
> We are planning to extend GEAR to additional modalities and domains in future work
>
> **2. Presentation and accessibility of the mathematical framework**
>
> We appreciate the concern that the differential-geometric formalism may be challenging for some readers. In the manuscript, we prioritized presenting a mathematically precise treatment of metrics and curvature, and we also included a pedagogical introduction for readers who may be less familiar with these concepts.
>
> That said, it is inherently difficult to convey both the explicit equations and the underlying meaning of differential geometry in a brief, introductory manner. The introductory material is intended to help motivated readers who wish to engage more deeply with the geometric foundations, but it cannot fully replace the formal mathematical framework.
>
> Because our method is fundamentally grounded in differential geometry, certain mathematical notions and derivations are unavoidable. We have aimed to balance rigor with readability, but simplifying the mathematics further would compromise the conceptual integrity of the approach.
>
> More broadly, scientific progress often involves the introduction of new or unfamiliar mathematical tools, which can naturally appear complex at first. As a research paper, the manuscript necessarily prioritizes correctness and clarity of the technical contributions rather than accessibility for a general audience.
>
> To support readers as much as possible, Appendices A and B introduce the necessary notation and definitions, while Appendix C provides the full derivation. Together, these materials are intended to help readers—including those new to these ideas—follow the key mathematical steps.
>
> **3. Two-task setting vs. multi-task extension**
>
> We agree that the current paper focuses on a two-task setting (one source, one target), and we appreciate the request to clarify how curvature matching scales to multiple tasks.
>
> Conceptually, GEAR extends naturally to multi-task scenarios in two ways:
>
> (1) Shared reference manifold (one “hub” frame).
> Define a universal latent frame $M_0$.
> For each task t, learn a task-specific transfer mapping $\Phi_t: M_t → M_0$.
> Apply metric and curvature losses between each task and the shared frame:
>
> $L_{geom} = \sum_t ( L_{metric(t→0)} + L_{curv(t→0)} )$.
>
> This avoids quadratic pairwise terms and scales efficiently.
>
> (2) Pairwise matching for small numbers of tasks.
> For a small set of tasks K, one may define pairwise alignment terms between selected task pairs known to be highly correlated.
> This generalizes our current two-task setup directly.
>
> In both cases, the only requirement is that latent dimensions match across tasks—already required in our two-task setting.
>
> We restricted the experiments to the two-task scenario because:
>
> - GATE, the baseline we compare against, is also evaluated exclusively in a two-task transfer setting, ensuring a fair and controlled comparison; and
>
> - Exploring the multi-task variant would require substantially more complex hyperparameter tuning and introduces several practical challenges that fall beyond the scope of this submission.
>
> However, we plan to extend the method to settings with more than two tasks in future work.

---

### Official Review · Reviewer_Lito · 2025-11-03

**Soundness:** 3
**Presentation:** 2
**Contribution:** 2
**Rating:** 4
**Confidence:** 3

**Summary:**

This paper proposes a new transfer learning (TL) framework, named GEAR (Geometric Embedding Alignment via cuRvature matching), designed for regression tasks, with a focus on molecular property prediction. The method interprets the latent spaces of deep learning models as Riemannian manifolds. Building upon prior work like GATE, which aligns latent spaces by matching local infinitesimal distances, GEAR aims to overcome the limitations of local alignment by enforcing a global geometric consistency. The core contribution is a novel loss term that encourages the Ricci scalar curvatures of the source and target latent spaces to match.

**Strengths:**

Originality: The primary conceptual contribution—using the matching of Ricci scalar curvature as an explicit objective for aligning latent spaces in transfer learning—is novel and intellectually stimulating. While the use of Riemannian geometry in deep learning is an active research area, the direct analytical computation and optimization of a global curvature property like the Ricci scalar for transfer learning(TL) is a creative  extension of existing ideas.

Significance: The paper addresses the important and practical problem of transfer learning in data-scarce domains, particularly for regression tasks in the molecular sciences. The empirical results presented are compelling. Achieving average RMSE improvements of 14.4% (random split) and 8.3% (scaffold split) over a strong predecessor like GATE is significant. If these results are robust and reproducible, the method could have a substantial impact on the field.

Quality & Effort: The technical effort invested in this work is substantial. The appendices contain extensive mathematical derivations, attempting to analytically compute the first, second, and third derivatives of the network's mapping to formulate the Ricci scalar curvature (e.g., Eq. 102, 107). The experimental evaluation is also comprehensive, comparing GEAR against seven different baselines across 23 task pairs and two different data splitting strategies (random and scaffold).

**Weaknesses:**

My main concerns with this paper revolve around its profound lack of clarity, questionable technical justifications, which collectively undermine the confidence in its impressive results.

1. Severe Clarity and Presentation Issues: The paper is exceptionally dense and challenging to parse, to the point of being almost impenetrable.

Overwhelming Equations: The core technical sections (Sec 3.2, Appendix C) present monolithic equations (e.g., Eq. 96, 102, 107) that span many lines. These are provided with minimal intuition or step-by-step explanation, making them hard to verify or build upon.

Confusing Notation and Logic: The logic behind key components is convoluted. For instance, the $l_{metric}$ loss (Eq. 19-21) is crucial for regularizing the common space, but its formulation and the accompanying text are opaque. The definition of ẑ′ is confusing, and the justification for how minimizing MSE against the identity matrix $η_ij$ enforces local flatness is absent.

Unexplained Complexity in Pseudocode: Algorithm 1 introduces an inner loop (for step k = 1, . . . , K) for iteratively computing the metric. This procedure is not mentioned or justified anywhere in the main text. This is a major omission that introduces unexplained computational complexity and makes the algorithm difficult to understand.
Inaccessible to the Target Audience: The paper fails to bridge the gap between differential geometry and transfer learning. The introduction immediately invokes terms like "diffeomorphism invariance" and "Ricci curvature" without providing the necessary high-level intuition for an audience expert in transfer learning but not necessarily in geometry. A reader should not have to be a specialist in both fields to grasp the core motivation of the paper. This failure in communication creates an unnecessarily high barrier to entry and signals poor exposition.


Why Ricci Scalar? The paper argues for global geometric alignment but settles for matching the Ricci scalar R. This scalar is a single number representing a highly simplified trace of the full curvature information contained in the Riemann tensor $R^i_{ljk}$. Two geometrically very different manifolds can share the same Ricci scalar at a point. The paper makes the strong claim that this is sufficient for "accurate alignment" without providing any justification or considering the trade-offs of not using the richer Ricci or Riemann tensors.

Motivation is Asserted, Not Proven: The central motivation is that local alignment methods like GATE are insufficient. However, this is presented as a given limitation rather than being demonstrated. There is no analysis or illustrative example showing a concrete scenario where GATE fails and GEAR succeeds due to its global perspective.

Lack of Justification for Complexity / Omission of Simpler Alternatives: The paper does not justify why the immensely complex machinery of Ricci curvature is the most parsimonious solution. The goal is to capture non-local, second-order properties of the latent space. A much simpler approach would be to align the second-order derivatives (i.e., the Hessians) of the transfer mappings directly. This would also capture the "bending" of the space but would be vastly simpler to compute (requiring only second derivatives of the network, not third) and implement. The authors provide no ablation or argument for why the significant leap in complexity to Ricci curvature is necessary or superior to such a simpler alternative.

Potential Weaknesses in Experimental Protocol:

Hyperparameter Complexity: The model has a complex loss function with five weighting hyperparameters (α, β, γ, δ, ϵ). Furthermore, Table 2 shows that the encoder architecture itself is a hyperparameter that varies for each of the 23 task pairs. The paper provides no details on how these numerous hyperparameters were tuned. This raises the concern that the strong performance may be the result of exhaustive, task-pair-specific tuning, which would limit the method's general applicability and make the comparison to baselines less controlled.
Unconventional Robustness Test: The noisy data experiment in Section 5.2 involves corrupting a subset of the test set, adding these corrupted points to the training set, and then evaluating on these same corrupted points. This is not a standard test of robustness to noisy labels. It primarily tests the model's ability to correct/memorize specific noisy examples it has seen during training, rather than its ability to generalize in the presence of a noisy training distribution.

**Questions:**

Questions


Regarding Algorithm 1: Please explain the purpose and motivation for the inner loop over k from 1 to K. Is this an iterative procedure performed at each training step? What is its computational cost, and why is it necessary?

Regarding $l_metric$ (Eq. 19-21): Could you provide a clear, step-by-step derivation and intuition for this loss term? The current explanation is difficult to follow. Specifically, how does the proposed formulation mathematically ensure that the manifold M is driven towards being locally flat?


Regarding the Choice of Curvature: Why did you choose to match the Ricci scalar R instead of a more informative quantity like the Ricci tensor $R_ij$? Since R is a trace, it loses significant geometric information. Have you explored matching tensor components, and if so, how did it affect performance and computational cost?

Regarding Hyperparameters: How were the loss weights (α to ϵ) and the 23 different task-specific encoder architectures (Table 2) selected? Was a systematic and consistent validation strategy employed? Please comment on the sensitivity of GEAR's performance to these choices.

Regarding Comparison to GATE: The motivation for GEAR rests on the purported limitations of GATE's local alignment. Can you provide a more direct and concrete analysis (e.g., a qualitative visualization of latent spaces for a specific task pair) that demonstrates a failure case of local alignment that is successfully addressed by your global curvature matching approach?

Regarding Method Complexity: Could you justify the use of Ricci curvature over simpler alternatives for capturing second-order geometric information? For instance, have you considered aligning the Hessians of the transfer maps? Please provide an argument or an ablation study showing that the added complexity of computing Ricci curvature is warranted by a significant performance gain over such simpler methods.

---

> ### Author Response · Authors · 2025-11-17
>
> Before reading the following comments, we kindly suggest that the reviewer refer to the summary above for a concise overview of our method.
>
> We thank the reviewer for the thorough reading and for acknowledging both the originality and potential impact of GEAR. Below we address the main concerns point-by-point and clarify the motivation and technical design choices that may not have been sufficiently emphasized in the main text.
>
> **1. On clarity, long equations, and accessibility**
>
> Long equations and Appendix C
>
> The very long expressions (e.g., Eqs. 96, 102, 107) were included to make the analytic derivations explicit. These equations are essential for deriving the main results of our work. Computing the Ricci scalar requires a detailed and sometimes tedious sequence of steps, and therefore we provided explicit computations together with index-notation explanations and a brief crash course on Riemannian differential geometry in Appendices A and B.
>
> We stated during submission that our work is grounded in differential geometry—specifically Riemannian geometry—both in the abstract and in the TL;DR. We therefore assumed that readers would have at least a basic understanding of the field. Terms such as “diffeomorphism invariance” and “Ricci curvature” reflect this context. Nevertheless, we made an effort to provide explanations of the relevant notation and mathematical background for readers who may not be specialists, which is not commonly done in scientific articles.
>
> **2. Algorithm 1 and the inner loop over k = 1,...,K**
>
> We apologize for the missing details. The inner loop over K indicates how many times the metric in the transfer module is updated. As the loop proceeds, the metric is repeatedly transformed under the diffeomorphism, and numerical error accumulates. Increasing K therefore places a stronger constraint on the metric, but also amplifies numerical sensitivity. For this reason, we keep K small (typically K = 2) in our experiments.
>
> The extra cost is dominated by Jacobian-related operations and is already reflected in our reported per-batch timing (≈0.50 s vs. ≈0.30 s for GATE/STL).
>
> We agree that Algorithm 1 currently over-emphasizes this loop without explaining its purpose. In the revision, we will:
>
> - Clarify that K is small and fixed
> - Add explanation about K loop in the main contents
>
> **3. Metric loss (Eqs. 19–21) and local flatness**
>
> Let f be the transfer module mapping a latent vector z to the common latent space. The Jacobian J = ∂f/∂z defines the pull-back metric g = Jᵀ J, which is the standard way to obtain a Riemannian metric from a smooth map. Here, g is the induced metric on the task-specific latent manifold.
>
> Our goal is to push this metric forward into the common manifold so that it becomes locally flat. To do this, we apply the Jacobian of the inverse transfer module, denoted J'. The pushed-forward metric is
> $\eta_{(s)} = J'^T g J'$.
>
> A locally flat (Euclidean) frame corresponds to $\eta_{(s)} \sim I$.
>
> Our metric loss is an MSE penalty enforcing this condition:
> metric_loss $ ∝ || \eta_{(s)} − I ||^2$.
>
> In our implementation (Eqs. 19–21), we construct local basis vectors and penalize deviations of their pairwise inner products from $\delta_{ij}$. This is equivalent to enforcing $\eta_{(s)} ≈ I$.
>
> Thus, minimizing the metric loss ensures that the learned local coordinate system behaves like an orthonormal Euclidean frame, which is the formal notion of local flatness.
>
> **4. Why Ricci scalar instead of Ricci/Riemann tensor?**
>
> The Ricci scalar curvature is a widely used representative geometric quantity in mathematics and physics. It is not simply the trace of the Ricci tensor or the Riemann tensor, but rather a contraction involving the fully curved, nonlinear metric tensor. As a result, the Ricci scalar is a complex nonlinear mapping from a vector to a single value that reflects the curvature around that point. In many settings, this quantity is sufficient to capture key geometric characteristics of the manifold.
>
> Moreover, the Riemann and Ricci curvature tensors are equivariant—not invariant—under diffeomorphisms. Comparing these tensors across different coordinate frames requires applying nontrivial and often tedious transformation rules. In contrast, the Ricci scalar curvature is both informative about the manifold’s geometry and invariant under coordinate transformations. This invariance makes it uniquely convenient and practical to use in applications where multiple coordinate systems are involved.

---

> ### Author Response · Authors · 2025-11-17
>
> **5. Motivation vs. GATE and the “local vs. global” distinction**
>
> We appreciate the request for a clearer comparison. GATE aligns tasks using local perturbations in a shared latent space and matching infinitesimal distances. This leads to several limitations:
>
> (1) Sensitivity to perturbation scale.
>
> A manually chosen perturbation radius must act as “infinitesimal.” But latent norms vary, making this radius either too small (no signal) or too large (leaving the local linear regime).
>
> (2) Local but not global alignment.
>
> Local neighborhoods may match, but their global arrangement can differ drastically (e.g., different curvature or manifold shape).
>
> (3) Shared embedding requirement.
>
> Because perturbations occur in a shared latent space, GATE requires source and target to share the same embedding, which is restrictive for multimodal or task-specific encoders.
>
> **How GEAR addresses these issues.**
>
> - We use task-specific encoders and align them in a shared geometric space defined by the transfer module.
>
> - We impose both metric and curvature constraints, shaping local and global geometry.
>
> - We eliminate sensitivity to arbitrary perturbation scales.
>
> **Empirical evidence.**
>
> - GEAR improves RMSE over GATE on 23 task pairs (Fig. 2, Appendix G).
>
> - Removing curvature in ablations increases overfitting and worsens performance.
>
> We agree that adding a qualitative visualization would be helpful; however, doing so is practically challenging. Both GATE and GEAR inherit geometric information from the underlying encoders, so the visible “shape” of the manifold does not differ in a clear or interpretable way. More fundamentally, there is no ground-truth or “correct” manifold shape to compare against. Our methods aim to align manifolds in a way that improves regression performance, but there is no canonical manifold to serve as a reference. As a result, visualizing differences meaningfully is nontrivial and can easily be misleading rather than informative.
>
> Instead, we provide extensive main experiments demonstrating the effectiveness of incorporating global geometry, as well as noise-corruption tests in the ablation studies, which together offer more reliable evidence than subjective visualizations.
>
> **6. Ricci vs. “simpler” second-order alternatives (e.g., Hessian alignment)**
>
> The suggestion to use Hessian alignment is based on a misunderstanding of the underlying geometry. There are several reasons why the Hessian is not an appropriate substitute for intrinsic curvature:
>
> (1) Geometric information is not contained in the Hessian.
> In Riemannian geometry, intrinsic curvature depends on third-order derivatives of the coordinate map (second derivatives of the metric), not on second derivatives of the function itself. The Hessian does not encode the manifold’s curvature structure.
>
> (2) Bases in curved spaces transform nontrivially.
> Unlike in flat Euclidean space, basis vectors on a curved manifold vary under coordinate transformations. Quantities like the Hessian depend heavily on the chosen coordinates and do not behave geometrically under diffeomorphisms.
>
> (3) Ordinary derivatives are not covariant.
> The Hessian is not a tensor unless one uses covariant derivatives. Because of this, there is no systematic or well-defined way to transfer or compare Hessians across different coordinate systems. Consequently, comparing Hessians from different tasks is not meaningful on equal footing.
>
> For these reasons, the Hessian is fundamentally unsuitable for our geometric alignment setting. The corresponding ablation is therefore mathematically trivial: it is expected not to work.
>
> Nevertheless, to demonstrate this empirically, we performed an experiment replacing the curvature loss with a Hessian-based loss. The validation RMSE curves clearly show that, unlike curvature, the Hessian loss fails to guide learning: performance is inferior and exhibits severe overfitting. This result is temporally added in the appendix H.
>
> As explained above, the Hessian does not provide a geometrically correct quantity for task-wise comparison, and thus we will not include this ablation in the final version of the paper.

---

> ### Author Response · Authors · 2025-11-17
>
> **7. Hyperparameters: loss weights and encoder architectures**
>
> We apologize for the previous lack of clarity.
>
> **Practical tuning strategy**
> We adopted a two-stage procedure:
>
> 1. Initialization with equal scale.
> All loss-weight hyperparameters were initialized to comparable magnitudes
> (𝛼 = 𝛽 = 𝛾 = 1, δ = ϵ = 0.1).
> This avoids early imbalance among loss components.
>
> 2. Grid search on validation sets.
> A grid over
> {0.1, 0.2, 0.5, 1} was used for α, β, γ, δ, ϵ.
> The optimal values were chosen based on average validation RMSE across 23 task pairs.\\
> This same procedure was applied consistently to all tasks.
>
> **Sensitivity analysis**
> We performed a full sensitivity sweep using the same grid.
>
> For the AS→target validation task:
>
> - Minimum RMSE: 0.3957
>
> - Mean RMSE across all combinations: 0.4314
>
> - Standard deviation: 0.0136
>
> - Ours (chosen configuration): 0.4033
>
> For ST → target validation task:
>
> - Minimum RMSE: 0.1746
>
> - Mean RMSE across all combinations: 0.1972
>
> - Standard deviation: 0.0141
>
> - Ours (chosen configuration): 0.1764
>
>
> Although the precise optimum varies per task, the selected setting is highly stable:
> it is near-optimal for AS and ST, and was empirically found to be the most robust across all 23 tasks.
> The variation across the grid is modest, indicating that GEAR does not rely on fragile or highly tuned hyperparameters.
>
> Finally, we emphasize that the complete ablation sweep for both target tasks required only ≈1400 seconds (under 30 minutes) on 8× A40 GPUs, demonstrating that the hyperparameter analysis is computationally lightweight and practical.
>
> **Conclusion**
> These results demonstrate that GEAR is stable under reasonable hyperparameter variations.
> We will include the AS, ST-task sensitivity table and a brief discussion of this analysis in the revised manuscript.
>
> Task-specific encoder architectures (Table 2).
> Architecture variations (e.g., depth, width) are intentionally small and follow:
>
> - The same DMPNN backbone family used in GATE,
>
> - Standard heuristics (slightly larger encoders for harder tasks or larger datasets).
>
> We do not perform bespoke, task-specific architecture searches. Instead:
>
> - We begin with a small predefined architecture (initially a 2-layer network with width [200, 200]) and explore a limited range (e.g., widths from 100 up to [200, 200, 200]).
>
> - We tune on a validation split using a unified protocol shared across both GEAR and baseline models.
>
> We clarified these details in Appendix F.2.
>
> **8. Noisy-label robustness experiment (Sec. 5.2)**
>
> We agree that our current experiment does not represent a standard noisy-label setting. Our intention was to perform a stress test, rather than model random label noise. This choice is motivated by practical considerations: in many scientific datasets, errors arise from systematic mistakes, not random flips. Examples include missing a minus sign or mixing units.
>
> For instance, a dataset may primarily record values in kcal, but some entries may mistakenly use kJ. Converting these requires multiplying by 4184, producing discrepancies orders of magnitude larger than the underlying values. Such structured, high-impact errors are common in real-world scientific data.
>
> Specifically, in our stress-test setup:
>
> - We corrupt labels for a subset of molecules,
>
> - Include these corrupted examples during training,
>
> - And evaluate the model on these same corrupted points.
>
> This setting tests memorization versus regularization, rather than classical robustness to random label noise.
>
> In the revision, we will:
>
> - Rephrase the section to avoid overstating robustness,
>
> - Clarify that our experiment evaluates targeted corruption stress, not random noise.
>
> **9. Summary**
>
> In summary, GEAR is not “GATE + one extra loss term,” but a shift from local, perturbation-based alignment to intrinsic, curvature-aware geometric alignment, supported by:
>
> - A metric loss enforcing local flatness,
>
> - A Ricci scalar loss regularizing global intrinsic geometry,
>
> - Task-specific encoders, and
>
> - Empirical evidence of improved transfer performance, including extrapolation and label corruption stress tests.

---

> > ### Comment · Reviewer_Lito · 2025-11-22
> >
> > We thank the authors for their detailed response. Can you explain more on the experiment of replacing the covariant Ricci with the simpler hessian?  I understand that Ricci is a covariant tensor,  and Hessian doesn't transform nicely under coordinate transform. But why does this covariance property matters for transfer learning tasks？

---

> ### Author Response · Authors · 2025-11-23
>
> We thank the reviewer for the swift follow-up question. Below we clarify (1) why covariance / invariance matters specifically in our transfer-learning setting, and (2) what exactly we observed when replacing the Ricci-based loss with a Hessian-based one.
>
> 1. Why does covariance / invariance matter for TL in GEAR?
>
> - Our core setting:
> Each task has its own encoder, so the latent coordinates for source and target are a priori unrelated. We only assume that they are two coordinate charts of the same underlying latent manifold. (detailed explanation can be found in the Appendix B) The transfer modules then learn diffeomorphisms (coordinate transformation rules) between these charts and the common manifold 𝓜. This is the fundamental geometric setup in Sec. 3 and Appendix. B–C.
>
> In this situation, a TL objective that compares geometric quantities across tasks should have the following property:
>
> - If we reparameterize either task’s latent space by any smooth bijection (i.e., change coordinates but keep the underlying function class and predictions unchanged), then the TL objective and its optimum should not change.
>
> Otherwise, the success or failure of transfer would depend on arbitrary choices of parameterization (e.g., how we scale / warp one encoder’s last layer), not on the underlying shared structure of the tasks.
>
> Ricci scalar R(g) is an intrinsic, diffeomorphism-invariant quantity: once the metric g is fixed, R(g) has the same value in any coordinate system. Changing coordinates via a diffeomorphism changes the metric components and the Christoffel symbols, but their combination in the curvature scalar cancels these changes. Thus:
>
> - If we pre- or post-compose an encoder with an invertible smooth map (reparameterize the latent), the predictions and the Ricci scalar at each point are unchanged after we re-express it with the new metric.
>
> - Matching Ricci scalars between tasks therefore defines an alignment objective that depends only on the geometry of each task’s latent manifold, not on how that geometry is written in coordinates. (This we call 'invariant')
>
> The Hessian ∂²f/∂z², in contrast, is not covariant on a curved manifold. It does not transform as a tensor under diffeomorphisms unless we replace ordinary derivatives by covariant derivatives and carefully account for the connection. In practice this means:
>
> - If we reparameterize the latent space by z̃ = ψ(z) (with ψ invertible), the Hessian of the same function expressed in z̃ can change drastically—even though the underlying map between manifolds is unchanged.
>
> As a result, an objective that tries to “match Hessians across tasks” depends strongly on the chosen coordinate systems of each encoder and transfer module. Two models that represent the same geometric alignment but use different parameterizations could incur very different Hessian losses.
>
> To make this concrete, imagine two people each create a map of the same city:
>
> - One draws the city using a Cartesian coordinate grid.
>
> - The other draws the city using a rotated and stretched grid.
>
> The two drawings look completely different, even though they represent the same physical city. If someone asked them to “align the maps” by matching the second derivatives of the grid lines (i.e., the Hessians), it would fail completely: stretching or rotating a map changes those derivatives, even though the underlying city remains unchanged.
>
> For transfer learning, this is exactly the failure mode we want to avoid: the cross-task coupling should not be about matching arbitrary parameterizations, but about matching the intrinsic geometry that governs how representations deform across tasks. Using a covariant / invariant geometric object (metric + Ricci curvature) ensures that:
>
> - The alignment loss is **coordinate-free**: it measures how “geodesics focus/spread” in each latent space, not how we numerically encode those geodesics.
>
> - The method is robust to benign architectural choices such as adding invertible linear layers, rescalings, or other smooth reparameterizations at the top of the encoder.
>
> - The comparison between tasks is meaningful even when their encoders are quite different (which is exactly the flexibility GEAR aims to provide).
>
> In short, the covariance/invariance property matters in TL because it guarantees that we are aligning what the latent spaces represent, not how we happened to parametrize them.

---

> > ### Author Response · Authors · 2025-11-23
> >
> > 2. What we did in the “Hessian instead of Ricci” experiment
> >
> > To address the reviewer’s suggestion more concretely, we implemented a variant of GEAR where we replaced the Ricci curvature loss with a Hessian-matching loss, keeping all other components (metric loss, mapping loss, consistency loss, architecture, and training protocol) unchanged.
> >
> > Construction of the Hessian loss.
> >
> > Let $f_s$, $f_t$ be the transfer maps for source and target tasks.
> >
> > For each batch, and for corresponding source/target latent points $z_s$, $z_t$ mapped to the common manifold, we computed the (ordinary) Hessians of the transfer maps, $H_s = \frac{\partial^2 f_s}{\partial z_s^2}$ and $H_t = \frac{\partial^2 f_t}{\partial z_t^2}$​, using autograd.
> >
> > We then defined a Frobenius-norm matching loss:
> > ${\cal{L}}_{Hess} = || H_s - H_t||^2_F$, scaled analogously to the curvature loss in the original GEAR.
> >
> > This directly mirrors the reviewer’s “simpler second-order alternative”: we align second derivatives of the transfer maps, not curvature.
> >
> > **Empirical behavior.**
> >
> > Across the tasks where we tested this variant (including those used in our ablation section):
> >
> > Training with ${\cal L}_{Hess}$ led to inferior validation RMSE compared to both the full GEAR model with Ricci curvature matching.
> >
> > Learning curves showed strong overfitting: the training loss kept decreasing while validation RMSE quickly plateaued or worsened, even under stronger regularization.
> >
> > We did not observe the kind of stabilization and overfitting suppression that appears when we include the curvature loss (as in Fig. 3 of the paper for the mapping+curvature vs. mapping-only setting).
> >
> > This behavior is consistent with the geometric argument above: the Hessian-based loss punishes differences that are largely artifacts of the chosen latent coordinates, so the optimizer spends capacity “fighting” coordinate choices instead of capturing the true shared structure between tasks. The Ricci-based loss, by contrast, regularizes a coordinate-free property of the metric and therefore provides a more stable and meaningful cross-task constraint.

---

### Author Response · Authors · 2025-11-17
**Brief summary of our work**

**Brief orientation for readers before the detailed responses**

We appreciate the reviewers’ time and would like to preface our replies with a short, self-contained overview of the mathematical and design choices behind GEAR, to ensure a common footing for the subsequent discussion.

**What GEAR is (and how it relates to GATE).**

GEAR is a methodological extension of GATE: it retains the transfer-learning structure but replaces GATE’s local, perturbation-based alignment with a curvature-guided alignment of latent spaces. This change is not a cosmetic loss tweak—it shifts the alignment objective from matching local distances to matching global geometric structure.

**Why Riemannian geometry is a natural model for latent spaces.**

Deep encoders built from smooth components (linear maps + smooth activations such as SiLU/Tanh/Sigmoid) define smooth maps; their image carries the structure of a smooth manifold, which can always be endowed with a Riemannian metric. Modeling latent spaces as Riemannian manifolds is therefore standard and mathematically justified. This does not require the input data or data-generation process to be smooth (e.g., discrete graphs are fine); smoothness of the network mapping is sufficient.

**Why curvature (in particular, Ricci) and locally flat frames.**

In curved spaces, plain (Euclidean) derivatives are insufficient; correct geometrical comparisons require covariant quantities derived from the metric and its connections. Curvature—computed from second-order derivatives of the metric—captures the intrinsic, coordinate-invariant shape of a manifold. The Ricci scalar provides a compact global summary of that shape, making it well-suited for aligning task manifolds. Using locally flat reference frames (via the induced metric/Jacobian) gives stable, unambiguous comparisons without hand-tuning perturbation scales, which was a limitation of GATE’s local-distance matching.

**Loss design and stability.**

GEAR’s objective separates concerns:

– Metric loss enforces local flatness of the reference frame;

– Curvature loss aligns global geometry;

– Mapping/consistency terms support cross-task transformation and prevent representational collapse;

– Supervised/reconstruction terms provide task signals.

We initialize loss weights ($\alpha, \beta, \gamma, \delta, \epsilon$) to comparable scales and tune them by grid search; training curves are smooth, indicating stable optimization.

**Reproducibility and scope.**

To stay within page limits, detailed architectures, dataset sizes, hyperparameters, and full tables (with uncertainties) are in the appendix; we will surface the most critical specs in the main text in revision. Although our experiments target molecular regression (the track we submitted to), the transfer module is modality-agnostic: any encoder (e.g., graph, vision, or language backbones) can feed GEAR, provided latent dimensions match.

With this shared context—Riemannian latent spaces, curvature-based alignment, the role of each loss term, and the scope of our evaluation—we address each reviewer’s questions directly below.

---

### Author Response · Authors · 2025-11-18
**Revised Version**

The revised manuscript has been uploaded, incorporating all changes requested by the reviewers. The updates are summarized below:

- Lines 317–322, 1380–1382: Clarified the role of the loop parameter 𝐾 in the main text and added further explanation in Appendix E.

- Lines 353–354: Added references to the backbone architectures used in our method to improve clarity.

- Lines 429–438: Rephrased Section 5.2 to avoid overstating robustness and clarified that the experiment evaluates targeted corruption stress rather than random label noise, which better reflects errors commonly occurring in practical scientific datasets.

- Figure 9 in Appendix H: Added a validation-curve comparison using a Hessian-based loss to demonstrate that the Ricci scalar is essential for stable performance. (This figure will be removed in the final camera-ready version.)

- Appendix H.2: Added an ablation study analyzing the selection of five distinct hyperparameters, demonstrating both the stability of the model with respect to these choices and the rationale behind the selected configuration.

---

### Note · Authors · 2026-01-27

**Comment:**

It is unfortunate that the assigned reviews indicate that the reviewers did not have sufficient background in differential and Riemannian geometry to assess our paper. We clearly state this basis in the title, abstract, and TL;DR, yet the reviews focused on elementary questions that can be resolved using standard textbooks rather than engaging with our main technical contributions.

We expected a more technical review process and constructive debate, but only one reviewer replied during rebuttal. The meta-review was also disappointing: even after we raised concerns about reviewer expertise and the quality of the feedback, the final decision largely repeated the reviewers’ statements without substantively addressing our points.

ChatGPT is certainly far better and far more professional…

**Withdrawal Confirmation:**

I have read and agree with the venue's withdrawal policy on behalf of myself and my co-authors.

---

### Meta-Review · Area_Chair_zsyW · 2026-01-06

**Summary:**

This paper introduces a curvature-based transfer learning framework that aims to align latent spaces by enforcing global geometric consistency. While the idea of leveraging differential geometry for transfer learning is conceptually interesting and the reported empirical gains on molecular property prediction tasks are non-trivial, several fundamental issues limit confidence in the contribution and its readiness for acceptance.

A major concern is the lack of clarity and accessibility. The core methodological sections are dominated by dense mathematical derivations with limited intuition, making it difficult for readers to understand the algorithmic pipeline or the practical role of each component. Key ideas are obscured by long, monolithic equations, and the connection between the theoretical formulation and the implemented losses is not clearly articulated. This significantly raises the barrier for comprehension, verification, and reuse.

The paper also suffers from insufficient justification of complexity. The proposed framework introduces a highly intricate loss design with multiple interacting terms and hyperparameters, yet provides little evidence that this level of sophistication is necessary. The work does not convincingly rule out simpler alternatives for capturing non-local or second-order structure, nor does it offer ablations or analyses that demonstrate why the chosen geometric formulation is essential rather than incidental to the observed performance gains.

There is a notable gap between theory and practice. While the motivation is framed around alignment on a shared underlying manifold, the practical implementation operates by matching curvature-related quantities computed independently in each latent space. The paper does not clearly explain under what conditions such matching guarantees meaningful global alignment, leaving the theoretical assumptions insufficiently grounded in the actual algorithm.

From an empirical perspective, although the evaluation is extensive within a single application domain, the claimed generality of the approach is not convincingly supported. Moreover, the absence of a systematic hyperparameter sensitivity analysis and clear tuning protocol raises concerns about robustness and reproducibility, particularly given the number of loss terms and architectural choices involved.

Overall, while the paper contains an interesting idea and encouraging experimental results, the combination of poor clarity, weak justification for methodological complexity, and unresolved theory–practice inconsistencies substantially undermines confidence in the contribution. Significant revision would be required to establish the method as a clear, robust, and broadly applicable advance.

**Reviewer Concerns:**

To reviewer Lito:

The rebuttal challenges the reviewer’s mathematical interpretation, but does not engage with the underlying concern regarding information loss in scalar curvature matching. As a result, the justification for using the Ricci scalar rather than higher-order curvature quantities remains unresolved.

---

To reviewer uBNt:

The rebuttal clarifies that the use of advanced mathematical tools such as differential geometry is principled and intentional, which addresses the reviewer’s concern at a high level regarding the legitimacy of the approach. However, the reviewer’s core concern was about accessibility and clarity of presentation, not about whether sophisticated mathematics should be used at all. The rebuttal does not explain how the exposition could be improved or how the method can be made more accessible to a broader audience. As a result, concerns regarding readability and clarity remain outstanding.

---

To reviewers oNjo and 6FvH

While the rebuttal correctly notes that the method includes additional loss terms beyond Ricci scalar matching, the reviewer’s core concern was about whether curvature-based matching, even with these additional constraints, is sufficient to achieve meaningful alignment onto a shared underlying manifold. This conceptual concern is not addressed by the rebuttal and therefore remains outstanding.

**Reviewer Scores:**

All the reviewers will keep their scores.

---

### Decision · Program_Chairs · 2026-01-26

Reject